# Antigen-selective modulation of AAV immunogenicity with tolerogenic rapamycin nanoparticles enables successful vector re-administration

Amine Meliani[1,2], Florence Boisgerault[2], Romain Hardet[1], Solenne Marmier[1], Fanny Collaud[2], Giuseppe Ronzitti[2], Christian Leborgne[2], Helena Costa Verdera[1,2], Marcelo Simon Sola[1,2], Severine Charles[2], Alban Vignaud[2], Laetitia van Wittenberghe[2], Giorgia Manni[3], Olivier Christophe[4], Francesca Fallarino [3], Christopher Roy[5], Alicia Michaud[5], Petr Ilyinskii[5], Takashi Kei Kishimoto[5] & Federico Mingozzi[1,2]

Gene therapy mediated by recombinant adeno-associated virus (AAV) vectors is a promising treatment for systemic monogenic diseases. However, vector immunogenicity represents a major limitation to gene transfer with AAV vectors, particularly for vector re-administration. Here, we demonstrate that synthetic vaccine particles encapsulating rapamycin (SVP[Rapa]), co-administered with AAV vectors, prevents the induction of anti-capsid humoral and cell-mediated responses. This allows successful vector re-administration in mice and nonhuman primates. SVP[Rapa] dosed with AAV vectors reduces B and T cell activation in an antigen-selective manner, inhibits CD8[+] T cell infiltration in the liver, and efficiently blocks memory T cell responses. SVP[Rapa] immunomodulatory effects can be transferred from treated to naive mice by adoptive transfer of splenocytes, and is inhibited by depletion of CD25[+] T cells, suggesting a role for regulatory T cells. Co-administration of SVP[Rapa] with AAV vector represents a powerful strategy to modulate vector immunogenicity and enable effective vector re-administration.

[1] Sorbonne Université and INSERM U974, 105 boulevard de l'Hôpital, 75651 Paris, France. [2] Genethon, UMR_S951 Inserm, Univ Evry, Université Paris Saclay, EPHE, 1 bis rue de l'Internationale, 91000 Evry, France. [3] Department of Experimental Medicine, University of Perugia, Piazzale Gambuli, 1, 06132 Perugia, Italy. [4] INSERM U1176 and Unité Mixte de Recherche S1176, Université Paris-Sud, Kremlin-Bicetre, 63 rue Gabriel Péri, 94270 Paris, France. [5] Selecta Bioscience, 480 Arsenal Street, Watertown, MA 02472, USA. These authors contributed equally: Amine Meliani, Florence Boisgerault. Correspondence and requests for materials should be addressed to F.M. (email: fmingozzi@genethon.fr)

Gene therapy mediated by recombinant adeno-associated virus (AAV) vectors is one of the most promising approaches for the treatment of a variety of inherited and acquired diseases[1]. Human clinical gene therapy trials with AAV have demonstrated durable expression at therapeutic levels when targeting tissues like the liver[2–7], motor neurons[8], and the retina[9]. Despite the exciting results to date, one of the limitations of the AAV vector gene transfer platform is the durability of the effect. For many metabolic and degenerative diseases, treatment is critically needed early in life[10,11], prior to the onset of irreversible tissue damage. However, because of their non-integrative nature, systemic gene therapy with AAV vectors in pediatric patients is expected to be limited by tissue proliferation associated with organ growth, which results in significant vector dilution over time[12–14]. Thus, maintaining the possibility to re-administer AAV is an important goal to achieve sustained therapeutic efficacy over time in pediatric patients. In addition, vector re-administration in both pediatric and adult patients would be desirable to enable vector titration, to increase the proportion of patients that achieve therapeutic levels of the transgene expression, while avoiding supra-physiological transgene expression[7] and potential toxicities associated with large vector doses[15].

However, vector immunogenicity represents a major limitation to re-administration of AAV vectors[16]. Persistent high-titer neutralizing antibodies (NAbs) are triggered following vector administration[5], which abolishes any benefit of repeated AAV-based treatments. In addition, experience in human trials has shown that induction of capsid-specific CD8[+] T cell responses can lead to clearance of AAV vector-transduced cells[3,5,6,17]. Thus, safe and effective strategies aimed at reducing AAV vector immunogenicity that allow for stable transgene expression and vector re-dosing are urgently needed.

Recently, administration of poly(lactic acid) (PLA) nanoparticles containing rapamycin[18,19] (SVP[Rapa]) has been shown to mitigate the formation of anti-drug antibodies when co-administered with protein therapeutics[18–23].

Here, we demonstrate that co-administration of SVP[Rapa] with AAV vectors induce safe and effective control of capsid immunogenicity in an antigen-selective manner. Importantly, this approach allows for productive repeated dosing of the same AAV serotype in mice and in nonhuman primates. Successful vector re-administration enabled by SVP[Rapa] allows for dose titration in the liver via targeting of additional populations of hepatocytes at each vector administration. In addition to inhibiting AAV-specific B cell activation, germinal center formation, and antibody production, SVP[Rapa] treatments also reduce antigen-specific T cell recall responses and prevent the appearance of CD8 T cell infiltrates in the liver. Our results suggest that SVP[Rapa] induces a population of regulatory cells that mitigate immune responses specific to the AAV serotype co-administered at the time of SVP[Rapa] treatment and are capable of transferring tolerance to naive recipients. Thus, SVP[Rapa]-mediated immunomodulation represents an attractive strategy to reduce AAV vector immunogenicity.

## Results

### SVP[Rapa] treatment allows for AAV vector re-administration.
To evaluate the ability of SVP[Rapa] to enable productive re-dosing of AAV vectors, male C57BL/6 mice were treated first with an AAV8 vector expressing luciferase (AAV8-luc) at day 0, followed by a second administration of an AAV8 vector encoding for human coagulation factor IX (AAV8-hF.IX) on day 21. In this setting, expression of the hF.IX transgene following the second injection of AAV is expected to be inhibited by the immune response induced by the first injection of AAV. Three experimental conditions were tested, (i) administration of both vectors with SVP[Rapa], (ii) administration of both vectors with SVP[empty] control, (iii) administration of the AAV8-hF.IX vector at day 21 only (no additional treatment control). Both AAV8-luc and AAV8-hF.IX vectors were infused at a dose of $4 \times 10^{12}$ vg kg$^{-1}$ (Fig. 1a). SVP[Rapa] inhibited the formation of anti-AAV8 IgG antibody responses after both the first and second injection of AAV vector (Fig. 1b). Conversely, animals treated with empty nanoparticles (SVP[empty]), and control naive animals infused with only $4 \times 10^{12}$ vg kg$^{-1}$ of an AAV8-hF.IX vector alone at day 21, developed high-titer anti-AAV IgG antibodies. Similarly, neutralizing antibody (Nab) titers, measured with an in vitro cell-based assay[24], were significantly higher in SVP[empty] control vs. SVP[Rapa]-treated animals (Fig. 1c). Accordingly, successful vector re-administration, measured by plasma levels of hF.IX transgene product, was achieved only in animals receiving SVP[Rapa] and in the control group of naive animals dosed at day 21 with AAV8-hF.IX vector only (Fig. 1d). Vector genome copy number (VGCN) in liver was consistent with plasma hF.IX levels, showing lack of transduction in livers of SVP[empty]-treated animals (Fig. 1e). VGCN was lower in SVP[Rapa] vs. naive animals treated at day 21 (Fig. 1e), possibly reflecting the presence of low titer anti-AAV NAbs which were not detected in the assays used. Importantly, co-administration of SVP[Rapa] at the time of each antigen exposure appeared to be critical for the inhibition of the anti-AAV humoral immune response, as re-administration of AAV8-hF.IX vector alone, with no SVP[Rapa], resulted in the induction of anti-capsid IgG response (Supplementary Fig. 1a, b). Accordingly, re-administration was not successful in animals not treated with SVP[Rapa] (Supplementary Fig. 2), and co-administration of SVP[Rapa] with AAV8-Luc was more effective in blocking anti-AAV antibodies than administration of SVP[Rapa] one day prior to vector administration (Supplementary Fig. 3).

We next evaluated the ability of SVP[Rapa] to enable a boost in transgene expression following a repeat dose of AAV vector administered three months after the initial treatment. C57BL/6 mice were dosed with an AAV8 vector encoding secreted human embryonic alkaline phosphatase (AAV8-SEAP) together with SVP[Rapa] or SVP[empty], then rested for 93 days prior to the second treatment (Fig. 1f). Mice treated with SVP[Rapa] showed no significant levels of anti-AAV8 IgG after two vector administrations (Fig. 1g). Importantly, SEAP transgene expression increased approximately twofold after the second injection (Fig. 1h). In contrast, control animals treated with AAV vector only or vector mixed with SVP[empty] showed a boost in antibody titers against AAV8 and no significant increase in SEAP transgene expression after the second injection (Fig. 1g, h). These results indicate that SVP[Rapa] nanoparticles, when co-administered at the same time of AAV vectors, efficiently block anti-capsid antibody responses, allowing for vector re-dosing and modulation of therapeutic efficacy.

### SVP[Rapa] permits AAV vector re-dosing in nonhuman primates.
We next assessed the activity of SVP[Rapa] administered in combination with AAV vectors in a pilot study with nonhuman primates (NHP, Macaca fascicularis), using a similar strategy as described in Fig. 1a. Three seronegative animals were treated with AAV8 vectors with either rapamycin nanoparticles (n = 2, SVP[Rapa]#1 and SVP[Rapa]#2) or SVP[empty] control (Fig. 2a). Two vectors encoding for secreted transgenes were administered to the animals, one at day 0 encoding for alpha-acid glucosidase (AAV8-Gaa) and one at day 30 encoding for hF.IX (AAV8-hF.IX.). Both vectors were given at a dose of $2 \times 10^{12}$ vg kg$^{-1}$. Animals that received the vector and SVP[Rapa] showed efficient

inhibition of anti-AAV8 antibody responses, both IgG and IgM, which lasted for the duration of the follow up (Fig. 2b, c). Accordingly, SVP[Rapa], but not control SVP[Empty] nanoparticles inhibited the formation of anti-AAV8 NAbs (Fig. 2d). This was confirmed with an in vivo neutralization assay[25,26] in

which naive mice were injected with AAV8-hF.IX together with serum from the experimental monkeys collected on day 30 prior to vector re-administration. Serum from the animal treated with SVP[empty], but not that from monkeys treated with SVP [Rapa], inhibited expression of hF.IX (Supplementary Fig. 4a).

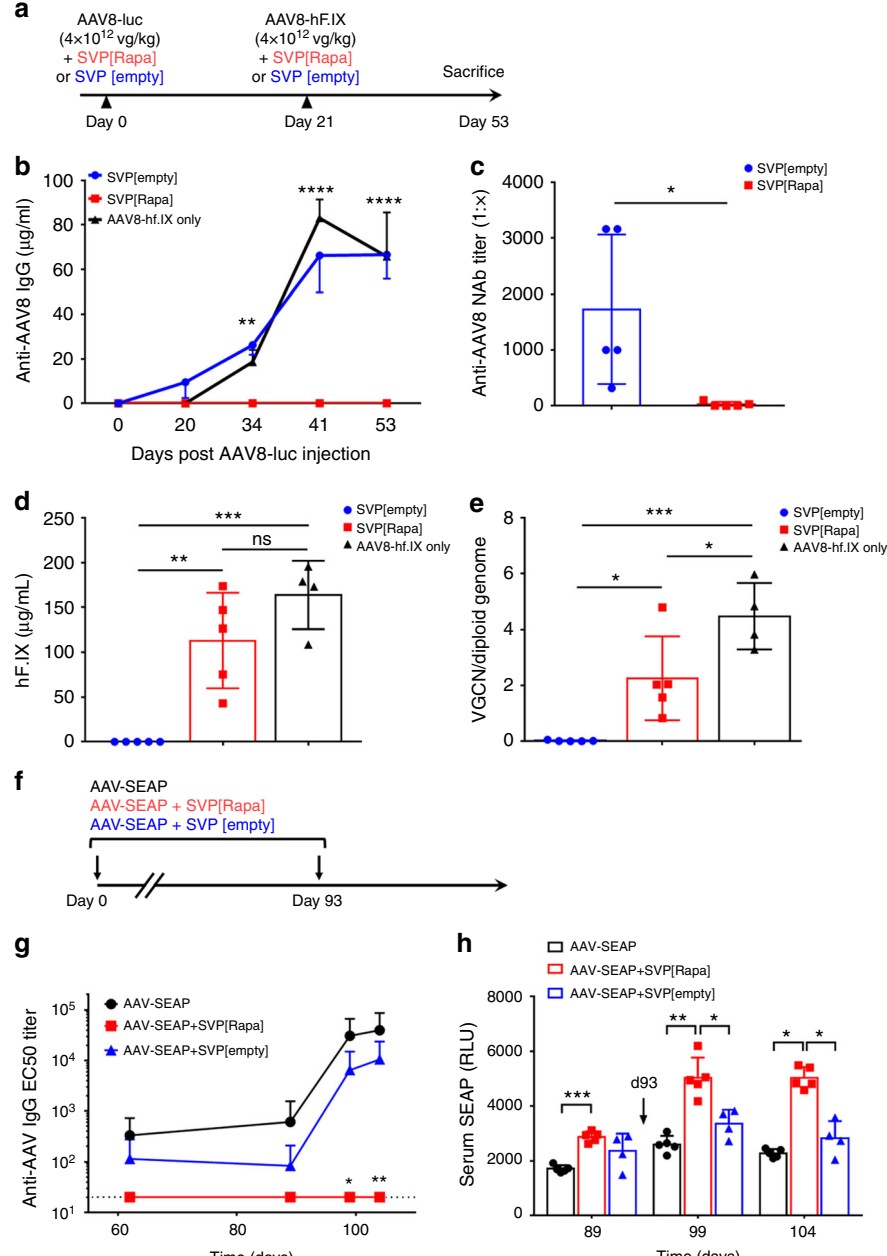

**Fig. 1** SVP[Rapa] treatment prevents anti-AAV capsid antibody responses in mice. **a** Protocol outline. Male C57BL/6 mice ($n = 5$ per group) were treated intravenously with $4 \times 10^{12}$ vg kg$^{-1}$ of AAV8-luc together with SVP[Rapa] (200 μg) or with SVP[empty] control. Three weeks later, animals were challenged with $4 \times 10^{12}$ vg kg$^{-1}$ of AAV8-hF.IX vector mixed with either SVP[Rapa] (200 μg) or SVP[empty] control. One additional control group ($n = 5$) of naive animals received $4 \times 10^{12}$ vg kg$^{-1}$ of AAV8-hF.IX vector alone at day 21 (defined as "AAV8-hF.IX only"). **b** Analysis of anti-AAV8 IgG antibodies measured by ELISA (**$p < 0.01$, ****$p < 0.0001$, refers to the significance between SVP[Rapa] and SVP[empty] or AAV8-hF.IX groups). **c** Anti-AAV8 neutralizing antibodies (NAbs) titers on day 54. **d** hF.IX antigen levels in plasma on day 54 determined by ELISA. **e** AAV8 vector genome copy number (VGCN) per diploid genome performed after killing (day 53). **f** Protocol outline. Three groups of C57BL/6 male mice ($n = 5$ per group) were prime-boosted (days 0 and 93, arrows) with $5 \times 10^{11}$ vg kg$^{-1}$ of AAV8-SEAP alone, or AAV8-SEAP mixed with SVP[Rapa] (50 μg), or AAV8-SEAP mixed with SVP [Empty] control. **g** Anti-AAV8 IgG antibodies measured by ELISA (*$p < 0.05$, **$p < 0.01$, refers to significance between SVP[Rapa] and AAV-SEAP). **h** Expressed SEAP activity in serum. Data are shown as mean ± s.d. Statistical analyses were performed by two-way ANOVA with Tukey post hoc test in **b**, **g**, and **h**, and by unpaired, two-tail *t*-test in **c** and by one-way ANOVA with Tukey post hoc test in **d** and **e** (*$p < 0.05$, **$p < 0.01$, ***$p < 0.001$, ****$p < 0.0001$; ns not significant)

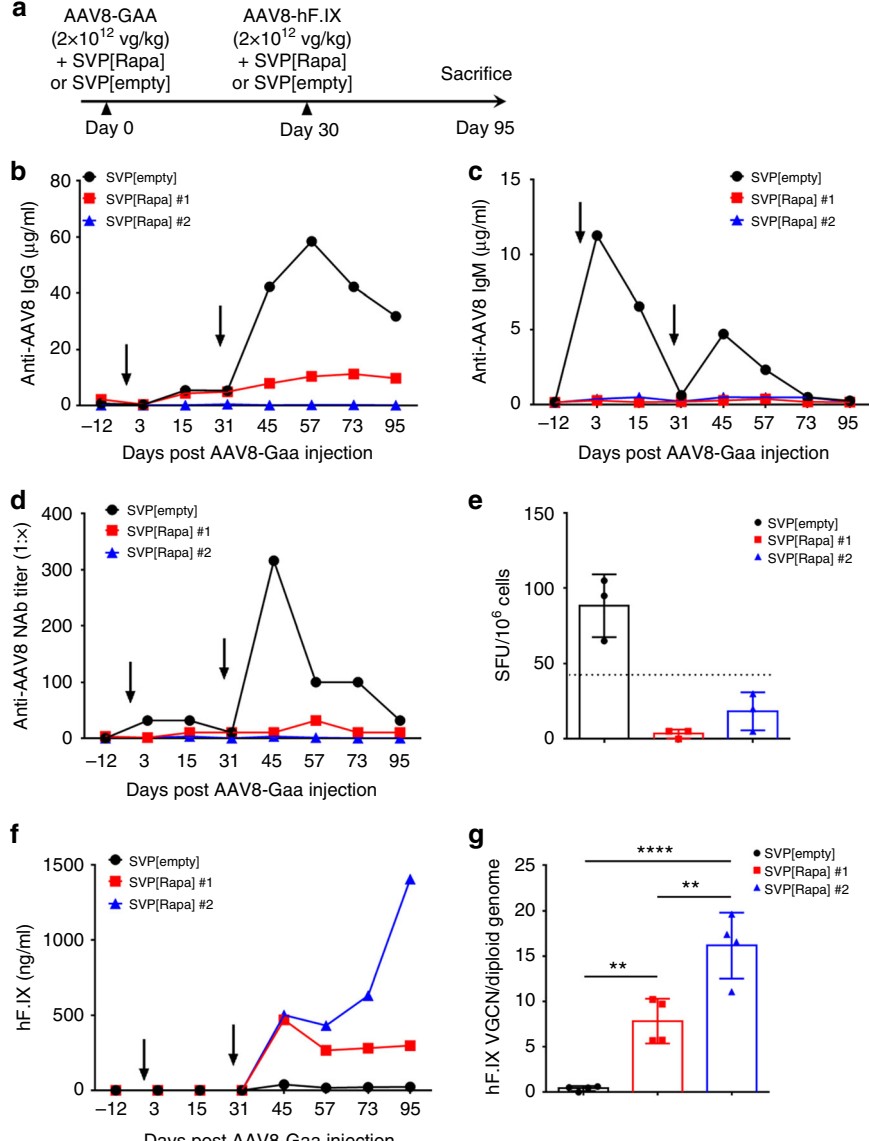

**Fig. 2** SVP[Rapa] treatment enables AAV8 re-administration in nonhuman primates. **a** Protocol outline. Male naive cynomolgus monkeys were treated i.v. with $2 \times 10^{12}$ vg kg$^{-1}$ of AAV8-Gaa vector and either SVP[Rapa] (3 mg kg$^{-1}$, n = 2, SVP[Rapa]#1 and SVP[Rapa]#2) or SVP[empty] control (n = 1) and then challenged i.v. on day 30 with $2 \times 10^{12}$ vg kg$^{-1}$ of AAV8-hF.IX vector and either SVP[Rapa] or SVP[empty] control, as described above. **b**, **c** Analysis of **b** anti-AAV8 IgG antibodies and **c** anti-AAV8 IgM antibody responses measured by ELISA. **d** Analysis of anti-AAV8 neutralizing antibodies (NAb) measured with a cell-based neutralization assay. **e** Analysis of anti-AAV8 IgG secreting B cells in splenocytes measured by B ELISpot. Data are shown as individual replicates and the bars represent mean ± s.d. The dotted line indicates the threshold for positivity corresponding to 50 spot forming units (SFU) per million cells. **f** Plasma hF.IX antigen levels quantified by ELISA at the indicated time points following administration of AAV8-hF.IX vector. **g** AAV8-hF. IX vector genome copy number (VGCN) per diploid genome in liver. The symbols represent individual liver lobes (left, right, caudate and quadrate) and the bars represent the mean ± s.d. (4 liver lobes per monkey; one-way ANOVA with Tukey post hoc test, **p < 0.01, ****p < 0.0001) The arrows represent the timing of each AAV8 vector infusion

AAV8-specific B cell ELISpot confirmed the suppression of antibody responses to the vector mediated by SVP[Rapa] (Fig. 2e). Consistent with the inhibition of anti-AAV8 humoral immune responses, vector re-administration with the AAV8-hF. IX vector resulted in detection of transgene product in plasma (Fig. 2f) and vector genomes in liver (Fig. 2g) only in SVP[Rapa]-treated animals. Interestingly, one animal, SVP[Rapa]#1, developed low but detectable IgG specific to AAV8 after the first vector administration (Fig. 2b) but showed no increase in antibody titers upon vector re-administration, demonstrating good control of primed B cells with SVP[Rapa]. SVP[Rapa] administration did not appear to have an impact on vector biodistribution

(Supplementary Fig. 4b). There was no alteration of clinical chemistry or hematology parameters in the SVP[Rapa] animals except for a transient increase in C-reactive protein around the time of treatment (Supplementary Tables 1–3).

These results suggest that inhibition of anti-AAV antibody formation mediated by SVP[Rapa] can be safely and effectively scaled up to a large animal model of gene transfer.

**SVP[Rapa] treatment enables enhanced hepatocyte targeting.** To investigate the pattern of vector transduction in liver parenchyma after repeated vector administrations, livers from NHPs

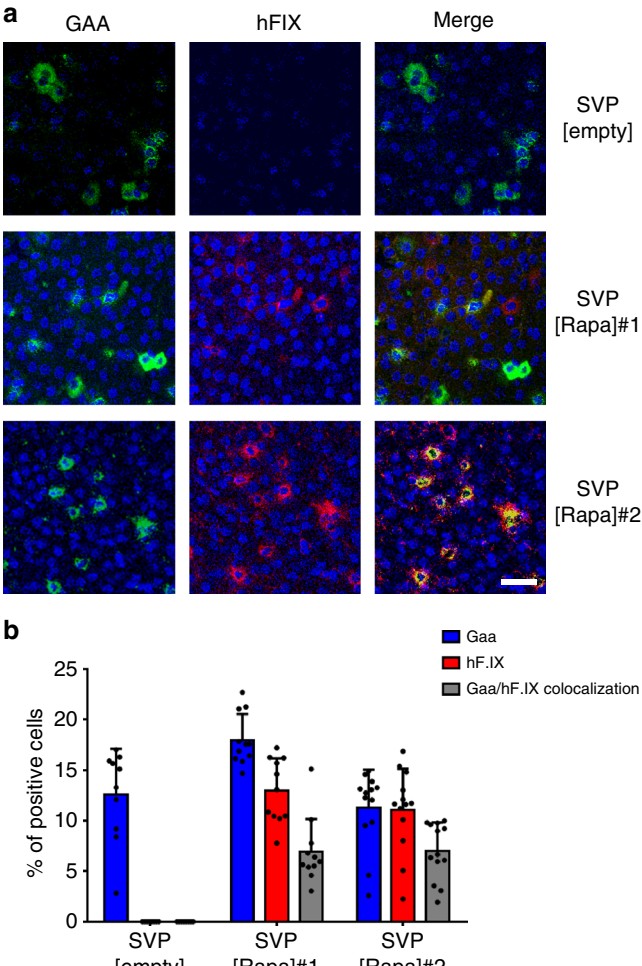

**Fig. 3** Enhanced liver transduction via repeated AAV8 vector administrations in nonhuman primates. **a** Dual immunofluorescence staining of Gaa and hF.IX in livers from nonhuman primates treated with AAV8 vectors together with SVP[Rapa] ($n = 2$, SVP[Rapa]#1 and SVP [Rapa]#2, 3 mg kg$^{-1}$) or SVP[empty] control ($n = 1$). 95 days post AAV8-Gaa administration, liver lobes were collected and stained for Gaa (shown in green), hF.IX (shown in red) and Gaa/hF.IX co-localization (merge, shown in yellow) and nuclei are DAPI-stained (Blue). Scale bar 20 µm. **b** Quantification of cells positive for Gaa, hF.IX and double-positives to Gaa/hF.IX in liver lobes. Data are shown as mean ± s.d. (four different images per lobe were analyzed for each animal)

treated with SVP[Rapa]/[empty] and AAV vectors (Fig. 2a) were biopsied, sectioned, and stained for the Gaa and the hF.IX transgene products. Hepatocytes positive for Gaa, the product of the first vector infused, AAV8-Gaa, were detectable in all 3 animals at similar levels (Fig. 3a, b and Supplementary Fig. 5a, b). Conversely, staining for hF.IX, derived from the AAV8-hF.IX vector administered at day 30, was detectable only in the SVP [Rapa] #1 and #2 animals but not in the control animal receiving SVP[empty] (Fig. 3a, b and Supplementary Fig. 5a, b). Quantification of hepatocytes from SVP[Rapa]-treated animals showed the presence of hepatocytes that were single positive for Gaa, single positive for hF.IX, and double positive for Gaa and hF.IX (Fig. 3a, b).

Similar experiments were conducted in mice with two vectors encoding for the non-secreted transgenes green florescent protein (AAV8-GFP) and human uridine diphosphate (UDP) Glucuronosyltransferase 1A1 (AAV8-hUGT1A1)[13], administered at day 0 and 21, respectively (Supplementary Fig. 6a). Liver sections

showed that the second transgene (hUGT1A1) was readily detectable in hepatocytes in animals receiving SVP[Rapa] at each vector administration compared to control animals which showed no double staining (Supplementary Fig. 6b, c).

These results indicate that repeated AAV vector administration enabled by SVP[Rapa] treatment can enhance liver gene transfer in part by increasing the number of hepatocytes that are transduced.

**Inhibition of humoral and cellular responses with SVP[Rapa].** Liver enzyme elevation and loss of transgene expression after AAV vector administration in humans correlates with the appearance of capsid-specific CD8$^+$ T cells in the peripheral blood[17]. We evaluated the liver of mice treated with AAV vectors and SVP[Rapa] or SVP[empty] (Fig. 1) for the presence of cells expressing *CD8* mRNA by quantitative RT-PCR. *CD8* mRNA was significantly elevated in control mice treated with SVP[empty]. In contrast, the level of *CD8* mRNA detectable in animals treated with AAV vector and SVP[Rapa] was not significantly different from that of naive mice (Fig. 4a). To further characterize the effect of SVP[Rapa] on antigen-specific B cells and T cells, C57BL/6 mice were immunized with an AAV8 vector encoding the VP1 structural protein of the AAV8 capsid (AAV8-VP1), an approach that has been shown to elicit robust humoral and cell-mediated immunity to the capsid[27]. Animals received the vector intravenously at a dose of $4 \times 10^{12}$ vg kg$^{-1}$, together with SVP [Rapa] or SVP[empty], and killed 14 days later. SVP[Rapa] treatment resulted in control of de novo AAV8-specific T cell reactivity as measured by interferon-gamma ELISpot using splenocytes from treated mice (Fig. 4b) and anti-AAV8 IgG and IgM responses, as measured by AAV-specific B cell ELISpot (Fig. 4c). Anti-IgG antibody subclasses analysis was consistent with these results, showing production of IgG1, IgG2a, IgG2b, and IgG3 only in SVP[empty]-treated control animals (Supplementary Fig. 7a). No change was observed in the frequency of total B220$^+$ B cells in spleens as measured by flow cytometry (Fig. 4d), while germinal center B cells in lymph nodes were decreased in SVP [Rapa]-treated animals (Fig. 4e, f), indicating that SVP[Rapa] inhibited B cell activation or differentiation. Frequency of monocytes (Supplementary Fig. 7b) and granulocytes (Supplementary Fig. 7c) in blood at day 14 post treatment was unchanged. Finally, flow cytometry analysis (Supplementary Fig. 8) of the frequency of CD4$^+$CD25$^+$FoxP3$^+$ regulatory T cells (Tregs, Fig. 4g), CXCR5$^+$PD1$^+$Foxp3$^+$ follicular regulatory T cells (Tfr, Fig. 4h), and CXCR5$^+$PD1$^+$FoxP3$^-$ follicular helper T cells (Fht, Fig. 4i) showed an increase in Tregs and Tfr, suggesting a role for these regulatory T cell subsets in SVP[Rapa]-mediated immunomodulation.

Next, we compared the ability of B cell depleting anti-CD20 antibodies[28,29] with SVP[Rapa] to modulate anti-AAV humoral responses (Supplementary Fig. 9a). Anti-CD20 antibody treatment efficiently depleted the B cells (Supplementary Fig. 9b), however, it failed to inhibit the anti-AAV8 IgG response following AAV vector administration (Supplementary Fig. 9c). Conversely, co-administration of SVP[Rapa] with the AAV8 vector blocked anti-vector antibody formation, independent of anti-CD20 treatment (Supplementary Fig. 9c), supporting the notion that efficient targeting of T helper cells through SVP [Rapa] treatment is more effective than directly targeting B cells through anti-CD20 depletion in preventing anti-AAV antibody responses.

**SVP[Rapa] control memory T cell responses to AAV.** As many humans are naturally exposed to wild-type AAV, and thus carry a pool of capsid-specific primed T cells[30,31], we evaluated the effect

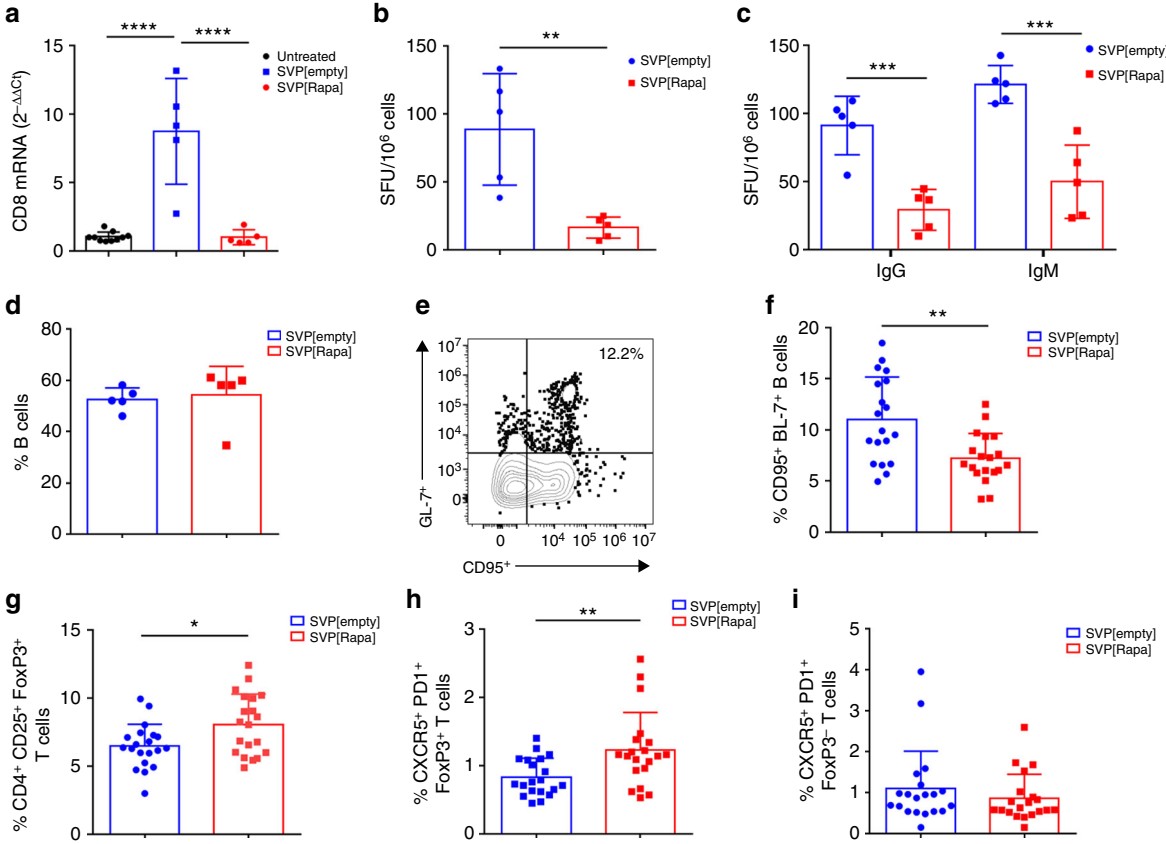

**Fig. 4** Inhibition of anti-AAV8 capsid cellular and humoral responses with SVP[Rapa] co-administration. **a** CD8 T cell infiltrates in the liver. Livers from animals treated in Fig. 1a were collected after killing on day 53 and evaluated for *CD8* mRNA expression by quantitative PCR using the $\Delta\Delta C_t$ method relative to housekeeping gene and to average of untreated mice. **b–d** Male C57BL/6 mice were treated with $4 \times 10^{12}$ vg kg$^{-1}$ of AAV8-VP1 vector together with SVP[Rapa] or with SVP[empty] control. 14 days later, spleens were collected for B and T cell assays. **b** Analysis of T cell recall responses after overnight stimulation with an AAV8 peptide pool in splenocytes measured by IFN-γ ELISpot and **c** anti-AAV8 IgG and IgM secreting B cell responses in splenocytes measured by B ELISpot. **d** Frequency of B220$^+$ CD19$^+$ B cells in spleens measured by flow cytometry. **e–i** Male C57BL/6 mice were treated with $4 \times 10^{12}$ vg kg$^{-1}$ of AAV8-luc vector together with SVP[Rapa] or with SVP[empty] control. 14 days later, animals were sacrificed. **e** Gating of germinal center (GC) B cells (CD95$^+$ GL7$^+$) are shown in representative flow cytometry plot. Cells were gated on B220$^+$IgD$^-$ cells. Shown is a mouse from the SVP[empty] control group. **f** Frequency of GC B cells (CD95$^+$ GL7$^+$) in lymph nodes of treated mice determined by flow cytometry as shown in **e**; **g** frequency of CD25$^+$ FoxP3$^+$ regulatory T cells in lymph nodes; **h** frequency of CXCR5$^+$ PD1$^+$ Foxp3$^+$ follicular regulatory T (Tfr) cells and **i** frequency of CXCR5$^+$PD1$^+$FoxP3$^-$ follicular helper T (Tfh) cells in lymph nodes. SVP[Rapa] treatment consisted of 200 μg of rapamycin. Data are shown as mean ± s.d. Statistical analyses were performed by one-way ANOVA with Tukey post hoc test in **a** and by unpaired, two-tail *t*-test in **b–d**, **f–i** ($n = 5$ in **a–d**, $n = 20$ in **f–i**). *$p < 0.05$, **$p < 0.01$, ***$p < 0.001$, ****$p < 0.0001$)

of SVP[Rapa] on memory immune responses. We first evaluated the durability of the effect of SVP[Rapa] on T cell reactivity to AAV capsid following repeated vector administrations. C57BL/6 mice ($n = 4$ per group) received an AAV8-Luc vector at a dose of $2 \times 10^{12}$ vg kg$^{-1}$ on day 0 and a second dose of $4 \times 10^{11}$ vg kg$^{-1}$ on day 77 with SVP[Rapa] or with SVP[empty] nanoparticles. T cell reactivity to the AAV capsid as measured by IFN-γ ELISpot at day 19 (after a single AAV dose) or day 80 (3 days after a repeat dose of AAV) was significantly lower in animals treated with SVP[Rapa] vs. SVP[empty] (Fig. 5a, b).

Next, we assessed the ability of SVP[Rapa] to control primed T cell responses following adoptive transfer of CD4$^+$ T cells from AAV-treated mice into naive mice (Fig. 5c). Donor animals were immunized with AAV-SEAP at a dose of $4 \times 10^{11}$ vg kg$^{-1}$. All donor mice developed antibodies to AAV8, indicating immune exposure (Fig. 5d). Sixty-two days later, animals were killed and $1 \times 10^7$ CD4$^+$ T cells isolated from spleens by negative selection were transferred to naive recipient mice which were then dosed with AAV8 vectors combined with SVP[Rapa] or SVP[empty] at day 1 and 21 post adoptive transfer. No anti-AAV antibodies were observed in SVP[Rapa]-treated recipient mice after the first

challenge of with $4 \times 10^{11}$ vg kg$^{-1}$ AAV8-RFP, and only low titer anti-AAV8 antibodies were observed in some animals at late time points after the second injection (Fig. 5e, diamond symbols). Conversely, all animals that received memory CD4$^+$ T cells but then treated with SVP[empty] developed high-titer AAV8 IgG antibodies (Fig. 5e, circles). These results suggest that SVP[Rapa] is capable of controlling re-activation of memory CD4$^+$ T cells upon re-exposure to the AAV capsid antigen.

**Antigen-selectivity and effect on Tregs of SVP[Rapa].** Immunosuppression can be associated with adverse events, including infections and malignancies. To test whether the control of immune responses to AAV vectors mediated by SVP[Rapa] was associated with general immunosuppression, we injected mice with an AAV8-Luc vector with SVP[Rapa] and then challenged them three weeks later, with the same capsid (an AAV8-hF.IX vector, Fig. 6b) or with a different capsid (an AAV5-hF.IX vector, Fig. 6c). An additional group of animals was challenged with human F.IX protein formulated in complete Freund's adjuvant (Fig. 6d). While animals challenged with the AAV8-hF.IX vector

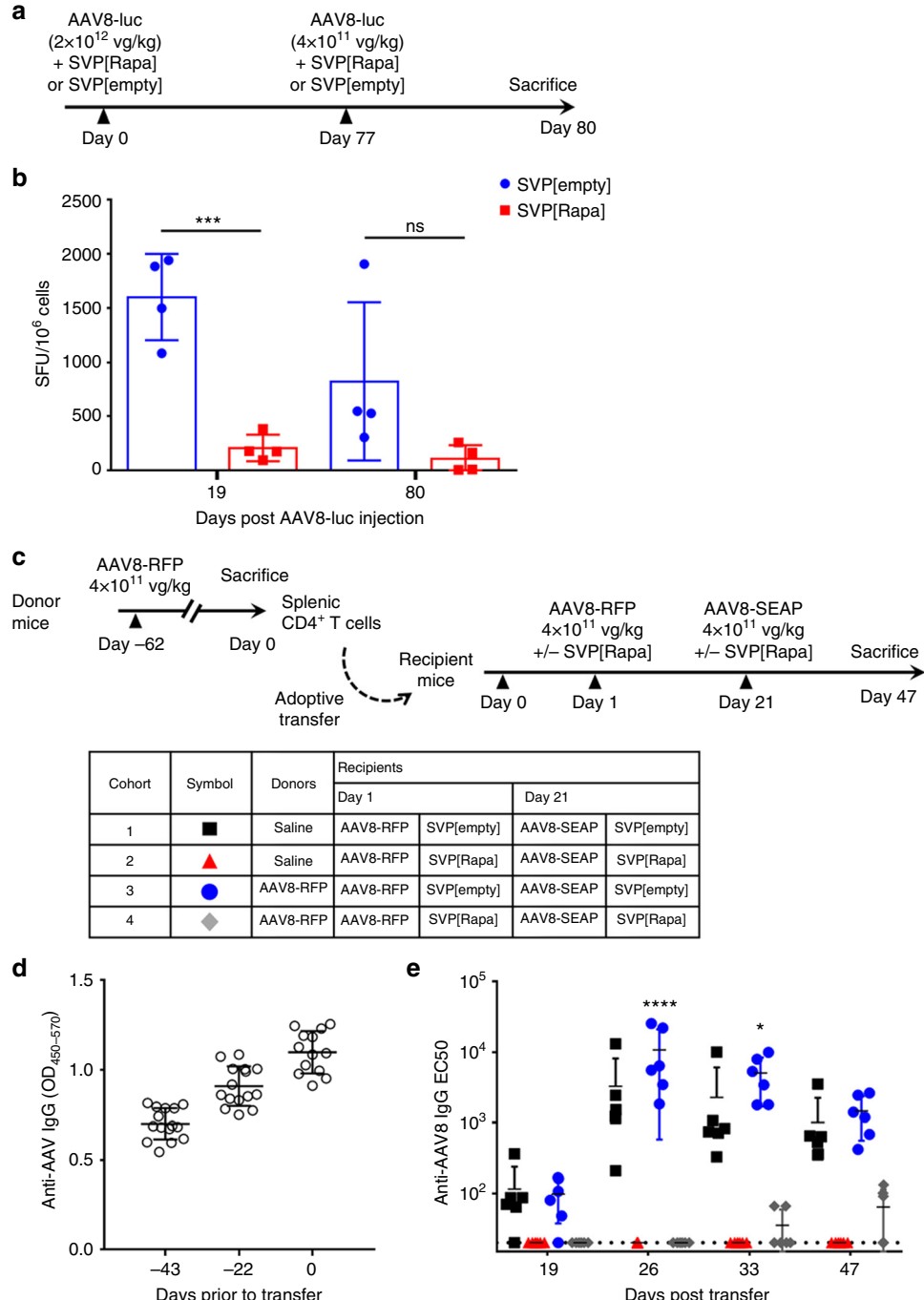

**Fig. 5** Control of anti-AAV8 capsid memory T cell responses with SVP[Rapa]. **a** Protocol outline. Male C57BL/6 mice ($n = 4$) were treated i.v. with AAV8-luc vector ($2.5 \times 10^{12}$ vg kg$^{-1}$) together with SVP[Rapa] or with SVP[empty] on day 0 and with a second dose on day 77 with AAV8-luc vector ($5 \times 10^{11}$ vg kg$^{-1}$) together with SVP[Rapa] or SVP[empty] control. Spleens were collected at day 19 and day 80 after priming and boosting, respectively. **b** IFN-γ ELISpot responses after 7 days of in vitro restimulation with an AAV8 peptide pool performed in splenocytes collected 19 or 80 days following initial vector injection. The symbols represent individual animals and the bars represent mean ± s.d. ($n = 4$, ***$p < 0.05$, ns not significant, two-tailed, unpaired $t$-test). **c** Protocol outline. Donor mice ($n = 15$) were left untreated or treated i.v with $4 \times 10^{11}$ vg kg$^{-1}$ of AAV8-RFP vector. 62 days later, spleens were collected and CD4$^+$ T cells were purified. $1 \times 10^7$ of CD4$^+$ T cells were then adoptively transferred into four groups of recipient mice ($n = 6$). Animals were treated with $4 \times 10^{11}$ vg kg$^{-1}$ of AAV8-RFP vector together with SVP[Rapa] or SVP[empty] control on day 1 post adoptive transfer and with $4 \times 10^{11}$ vg kg$^{-1}$ of AAV8-SEAP together with SVP[Rapa] or SVP[empty] on 21 post adoptive transfer. The tabular legend represents the summary of the treatment groups and the symbols represent the different cohorts of recipient mice. **d**, **e** Analysis of anti-AAV8 IgG antibodies in donor mice (**d**) and in recipient mice post adoptive transfer (**e**) measured by ELISA. **d**, **e** Data shown as mean ± s.d. ($n = 6$, *$p < 0.05$, ****$p < 0.0001$, refers to significance between cohorts 3 and 4 as determined by two-way ANOVA with Tukey post hoc test). SVP[Rapa] treatment consisted of 50 µg of rapamycin

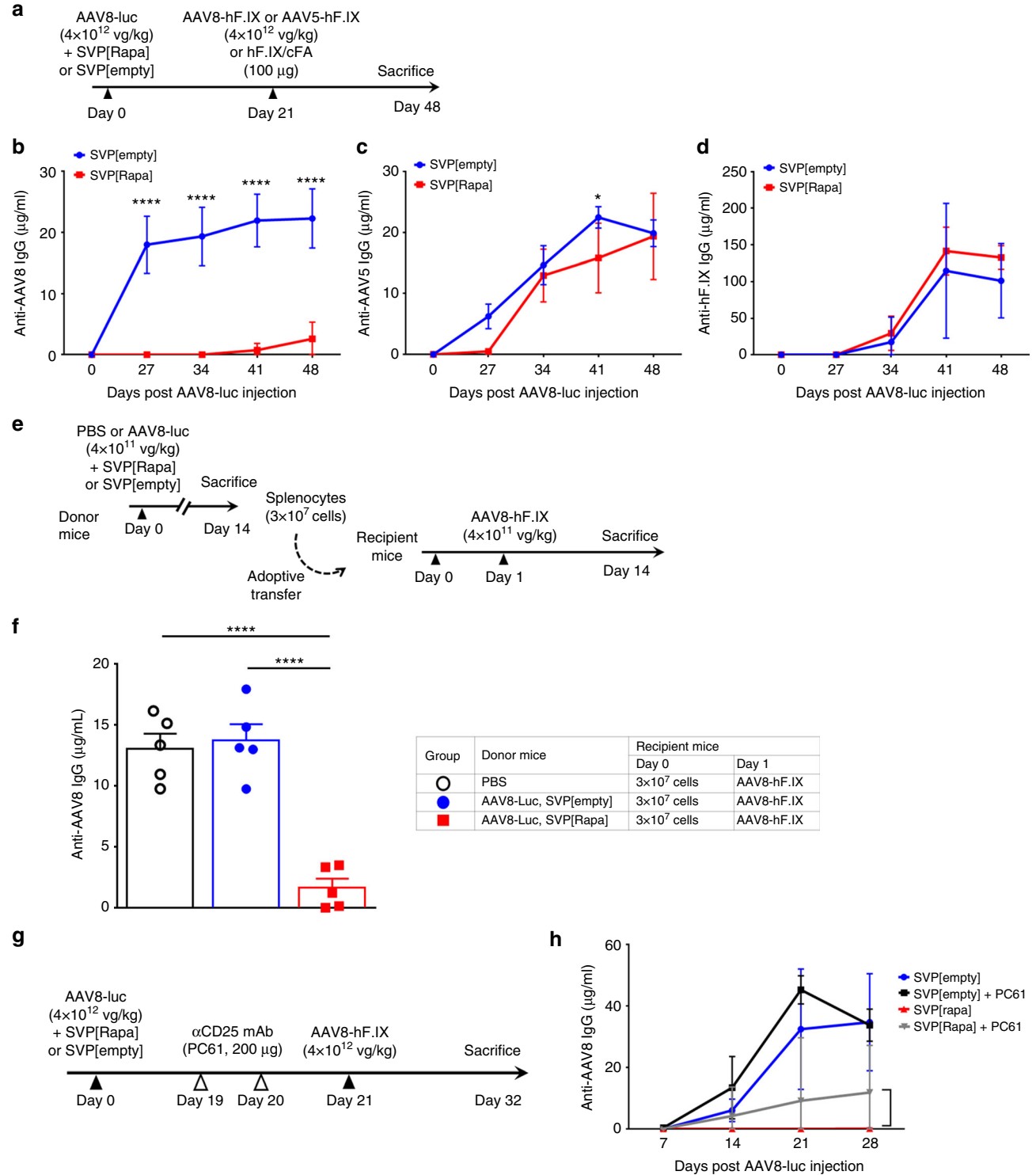

showed good control of humoral responses against the AAV8 capsid (Fig. 6b), animals developed a robust IgG antibody response against the AAV5 capsid (Fig. 6c) and hF.IX protein (Fig. 6d). These results indicate that the immunomodulatory effect of SVP[Rapa] is limited to the antigen co-administered with the nanoparticles. As antigen-specificity could reflect immune tolerance mechanism(s), and because upregulation of regulatory T cells was observed in lymphoid organs of SVP[Rapa] treated animals (Fig. 4f, g), we hypothesized that Tregs had a role in the immunomodulatory effect of SVP[Rapa]. One of the hallmarks of

antigen-specific tolerance is the ability to transfer tolerance from tolerized mice to naive mice. Accordingly, splenocytes from donor mice treated with AAV8-luc vector alone or together with SVP[Rapa] or SVP[empty] were collected 14 days post treatment and adoptively transferred into recipient mice, which were then challenged one day later with AAV8-hF.IX vector (Fig. 6e). Cells from donor mice treated with AAV8-luc vector co-administered SVP[Rapa] were able to suppress anti-AAV8 IgG response in recipient mice after challenge (Fig. 6f). Conversely, robust anti-capsid IgG formation was observed in mice that received cells

**Fig. 6** Antigen-specificity of SVP[Rapa] treatment and role of regulatory T cells. **a** Protocol outline. Male C57BL/6 mice ($n = 5$) were first injected i.v. with $4 \times 10^{12}$ vg kg$^{-1}$ of AAV8-luciferase vector together with SVP[Rapa] or with SVP[empty] control. Animals were then challenged i.v. on day 21 with $4 \times 10^{12}$ vg kg$^{-1}$ of AAV8-hF.IX vector or AAV5-hF.IX vector or injected s.c. with hF.IX in complete Freund's adjuvant (cFA). **b** Anti-AAV8 IgG in animals primed with AAV8-luc and challenged with AAV8-hF.IX. **c** Anti-AAV5 IgG in animals primed with AAV8-luc and challenged with AAV5-hF.IX, and **d** anti-hF.IX IgG antibodies determined by ELISA in animals primed with AAV8-luc and challenged with hF.IX protein in cFA. **e** Protocol outline. Donor mice ($n = 5$) were left untreated or treated i.v. with $4 \times 10^{11}$ vg kg$^{-1}$ of AAV8-luciferase vector together with SVP[Rapa] or SVP[empty] control. 14 days later, spleens were harvested and $3 \times 10^{7}$ splenocytes were adoptively transferred into recipient mice ($n = 5$). All animals were then treated one day later with $4 \times 10^{11}$ vg kg$^{-1}$ of AAV8-hF.IX vector. **f** Anti-AAV8 IgG antibodies measured 14 days post AAV8-hF.IX vector injection and measured by ELISA. **g** Protocol outline. Male C57BL/6 mice ($n = 5$ per group) were treated i.v. with $4 \times 10^{12}$ vg kg$^{-1}$ of AAV8-luciferase vector together with SVP[Rapa] or SVP[empty] control on day 0, followed by Tregs depletion (anti-CD25, clone PC61, 200 μg per mouse) on days 19 and 20. All animals were then challenged with $4 \times 10^{12}$ vg kg$^{-1}$ of AAV8-hF.IX vector alone on day 21. **h** Analysis anti-AAV8 IgG antibodies measured by ELISA. SVP[Rapa] treatment consisted of 200 μg of rapamycin. Data are shown as mean ± s.d. Statistical analyses were performed by two-way ANOVA with Tukey post hoc test in **b**–**d**, by one-way ANOVA with Tukey post hoc test in **f**, and by non-parametric unpaired, two-tail $t$-test in **h** (*$p < 0.05$, **$p < 0.01$, ***$p < 0.001$, ****$p < 0.0001$)

from donor mice primed with AAV8-luc and SVP[empty] control (Fig. 6f). To further confirm the role of Tregs cells in SVP[Rapa] control of vector immunogenicity, we performed a depletion experiment using an anti-CD25 antibody (clone PC61) to deplete Tregs. Animals were primed at day 0 with an AAV8-Luc vector (Fig. 6g). Before AAV8-hF.IX vector re-administration at day 21, animals received two injections of the antibody PC61 at day 19 and 20 (200 μg per dose). PC61 administration resulted in profound depletion of CD4$^+$CD25$^+$ T cells, as measured in spleen and lymph nodes after killing (day 32, Supplementary Fig. 10). Treg depletion led to a partial restoration of anti-AAV8 IgG antibody responses in SVP[Rapa] treated animals compared to animals treated with SVP[Rapa] with no Treg depletion (Fig. 6f). Conversely, PC61 administration together with SVP[empty] control had no effect on anti-AAV antibody levels. Together these findings suggest that Tregs have a role as mediators of SVP[Rapa] immunomodulatory activity, although other factors may contribute to the observed complete control of capsid immunogenicity.

## Discussion

Mitigating the immunogenicity to AAV is particularly challenging because of its size, the repetitive display of antigenic epitopes on the capsid[32,33], and the high degree of antibody suppression required to prevent vector neutralization[3,34]. Our results indicate that SVP[Rapa] has the potential to be a safe and effective approach to control capsid immunogenicity associated with systemic delivery of AAV vectors. SVP[Rapa] prevented the formation of NAbs and enabled successful vector re-administration in mice (Fig. 1) and nonhuman primates (Fig. 2) in an antigen-selective manner (Fig. 6b). Furthermore, we demonstrated that repeat dosing of AAV vectors with SVP[Rapa] can be used to increase expression of a transgene for example in the liver via targeting of additional hepatocyte populations at each vector infusion (Fig. 3 and Supplementary Fig. 5). These findings are of high relevance to gene therapies for diseases with early lethality[10], in which early intervention may require subsequent treatments to maintain efficacy over time[12]. They also address the need for strategies to carefully dose-titrate AAV vectors to avoid overexpression[7], or to target systemic diseases[35] in which multiple vector administrations are likely needed to achieve full therapeutic efficacy[36].

Despite numerous attempts, the ability to re-administer AAV vector still represents an unmet goal for the field of in vivo gene transfer. Some degree of success has been achieved in the context of intramuscular gene transfer[37] or gene transfer to immuno-privileged body sites[38,39], in which vector neutralization by circulating antibodies is not a concern[40].

The goal of addressing AAV capsid immunogenicity is not limited to prevention of antibody responses to allow for vector

administration. Results in human trials indicated that induction of CD8$^+$ T cell responses directed to the AAV capsid correlates with liver inflammation and can limit the duration of transgene expression[2–7,17]. While administration of corticosteroids has been effective in blocking these responses in some cases[2,5,6], in other instances expression was lost despite immunosuppression[41,42], indicating that more effective strategies need to be developed. Here, we observed a marked suppression of de novo capsid T cell responses with SVP[Rapa], even in the context of the expression of the capsid VP1 protein[27] (Fig. 4b). Although liver enzyme elevation is not typically observed in animal models of AAV-mediated gene therapy[17,43], we did observe evidence of CD8 T cell infiltrates in the liver after vector administration, which were inhibited by SVP[Rapa] treatment (Fig. 4a). Additionally, we observed inhibition of memory T cell responses, in a setting closer to that of gene transfer in humans pre-exposed to the virus[30,31] (Fig. 5). Thus, despite the limitations of animal models in predicting anti-capsid CD8$^+$ T cell responses[17,43], our results suggest that administration of SVP[Rapa] may be beneficial in terms of modulation of both humoral and cell-mediated responses to the AAV capsid.

Immunosuppression, including the use of free rapamycin in combination with other immunosuppressive agents[44], has been tested in gene therapy to prevent anti-vector antibody formation. This approach has proved poorly efficacious in preventing anti-AAV induction, even when potent drug combinations were used[45], in some cases resulting in detrimental interference with the induction regulatory T cells, which are essential for transgene maintenance in liver gene transfer[44,46]. Recently, a clinical case study was published on the effect of long-term depletion of B cells using rituximab combined with rapamycin and steroids administration around the time of AAV vector administration, which resulted in the abrogation of the anti-capsid antibody response[28]. However, data available indicate that long-term immunosuppressive treatment is likely needed in order to be efficacious. Our results indicate that transient depletion of B cells with an anti-CD20 antibody had no substantial effect on the anti-capsid antibody response in mice and provided no additional benefit to SVP[Rapa] treatment (Supplementary Fig. 8).

Plasmapheresis has also been proposed to re-administer AAV vectors[47]; however, this maneuver has several limitations, as human studies indicate that repeated sessions will be needed to eliminate anti-AAV antibodies present in the extravascular space, and that this approach is unlikely to completely eradicate high-titer antibodies[48]. Similarly, capsid switching has been proposed[49], however this is also a very complex procedure from a drug development and manufacturing perspective, and it is further complicated by high degree of cross-reactivity of anti-AAV antibodies[50].

In the setting of AAV liver gene transfer, administration of SVP[Rapa] concomitant to each vector exposure was sufficient to

completely inhibit anti-capsid immune responses. Co-administration of SVP[Rapa] with each dose of AAV vector appeared to be essential to completely block anti-AAV antibody responses, as repeated vector administration with no SVP[Rapa] resulted in antibody formation even when SVP[Rapa] was co-administered with the first vector dose (Supplementary Fig. 1). Consistently, data generated indicate that the effect of SVP[Rapa] is selective for the co-administered antigen, as we showed that there was no effect on the antibody response to a different AAV serotype or to an unrelated protein antigen administered at a later time point (Fig. 6b). Regulatory T cells seem to be involved as suppressors of capsid immunogenicity mediated by SVP[Rapa], as previously shown for other antigens[18,19] and as demonstrated by the observed increase in Treg frequency in lymphoid organs of treated mice (Fig. 4g), and by adoptive T cell transfer (Fig. 6f) and Treg depletion experiments (Fig. 6h). These observations are consistent with the published beneficial effect of rapamycin on the homeostasis of Tregs[51], and further supports the safety of the approach, as SVP[Rapa]-induced immune modulation appears to be highly antigen-selective, and because Tregs require antigen persistence for homeostasis[52]. The inability of anti-CD25 treatment to completely restore the anti-AAV antibody response may reflect incomplete deletion of CD25+ T cells in lymphoid organs or involvement of CD25-negative regulatory cells[53]. Alternatively, it is plausible that other mechanisms are mediating the effect of SVP[Rapa] on capsid immunogenicity, including the immuno-suppressive effect of rapamycin. Whether the effect of SVP[Rapa] observed here, in the context of liver gene transfer with AAV8 vector doses that are clinically relevant[5,6], will be identical for other capsids or at higher vector doses remains to be established. Future studies in larger cohorts of nonhuman primates will be required for the clinical translation of the results shown here.

We have previously demonstrated the functional benefit of using SVP[Rapa] to mitigate the formation of anti-drug antibodies (ADAs) against a variety of biologic drugs, including coagulation factor VIII in a model of hemophilia A[23], anti-TNF monoclonal antibody in a model of spontaneous arthritis[18], acid alpha-glucosidase in Pompe disease mice[20], immunotoxins in a model of mesothelioma[21], and pegylated uricase in uricase-deficient mice and in nonhuman primates[18]. Additionally, the safety and ability of SVP[Rapa] to inhibit the formation of ADAs against pegsiticase, a highly immunogenic enzyme therapy, in a dose-dependent manner was demonstrated in a phase 1 clinical trial (Clinicaltrials.gov NCT02648269)[22]. SVP[Rapa] is currently being evaluated in combination with pegsiticase in an ongoing multi-dose phase 2 clinical trial (Clinicaltrials.gov NCT02959918)[22].

One open question about the work presented here is whether SVP[Rapa] administration will impair the ability of the immune system to fight infections. Because of the need for SVP[Rapa] to be co-administered with the antigen, in order to guarantee uptake from the same subset of antigen presenting cells[18,21], it is less likely that SVP[Rapa] will lead to global immunosuppression, although future studies are needed to carefully address this point. The lack of immune cell depletion observed in animals treated with AAV vectors and SVP[Rapa], both rodents and nonhuman primates, provides additional data supporting the safety of the approach. Initial clinical data in small numbers of patients treated with SVP[Rapa] are consistent with these results, as they do not show any increased risk of infection (Clinicaltrials.gov NCT02648269; NCT02959918)[22]. Although AAV is considered to be a non-pathogenic virus and is not associated with any disease, it is unknown if there would be any consequences of inducing immune tolerance to AAV. Further evaluation of safety will have to be addressed in preclinical studies and clinical trials. Finally, another question about the technology is whether it could be used to eradicate established antibody responses to AAV. SVP

[Rapa] has been shown to reduce antibody titers against protein antigens in primed animals[21,23]. However, since very low titers of neutralizing antibodies can inhibit AAV transduction, it is unclear if a reduction in anti-AAV titer would be sufficient to enable productive transgene expression, particularly in the presence of long-lived plasma cells. We are investigating whether SVP[Rapa] combined with drugs targeting B cells or plasma cells[54] may enable vector administration in the setting of pre-existing neutralizing antibodies.

In conclusion, the work presented here addresses AAV vector immunogenicity and specifically the longstanding challenge of vector re-administration. SVP[Rapa] co-administration with AAV vectors is a powerful, yet highly specific technology that enables the prevention of anti-AAV humoral immune responses and T cell reactivity to the capsid. This is a promising approach for safe and effective repeated vector dosing to dose titrate the therapeutic effect and to re-treat recipients of gene therapy in case efficacy is lost over time, thus opening new therapeutic avenues for AAV vector-mediated gene transfer for diseases requiring systemic transduction[35] or treatment in childhood due to early lethality[10].

## Methods

**SVP[Rapa] nanoparticles production.** Rapamycin-loaded poly(lactic acid) (PLA) nanoparticles (SVP[Rapa]) were prepared using oil-in-water single-emulsion evaporation method[18,19]. Briefly, rapamycin, PLA, and PLA-PEG block copolymer were dissolved in dichloromethane solution to form the oil-phase. The oil-phase was then added to an aqueous solution of polyvinylalcohol in phosphate buffer followed by sonication (Branson Digital Sonifier 250A, Branson Ultrasonics, Danbury, CT). The emulsion thus formed was added to a beaker containing phosphate buffer solution and stirred at room temperature for 2 h to allow the dichloromethane to evaporate. The resulting nanoparticles containing rapamycin were washed twice by centrifugation at 76,600×g at +4 °C and the pellet was resuspended in phosphate buffer solution. The bare nanoparticles without rapamycin (SVP[empty]) were prepared in identical conditions. Rapamycin content was measured by first transferring the nanoparticles into an organic solvent, followed by analysis via reverse-phase HPLC using a water/acetonitrile gradient containing 0.1% TFA. Typically, SVP[Rapa] contained ~10% rapamycin load; in a given preparation of SVP[Rapa], the concentration of rapamycin was adjusted to 2 mg ml. Nanoparticle size was ~200 nm as determined by dynamic light scattering. Each SVP[Rapa] treatment consisted of 50 or 200 µg of rapamycin as indicated in figure legend.

**AAV vector production.** AAV vectors were produced using adenovirus-free transfection method and purified by CsCl gradient centrifugation[55,56]. Genome-containing AAV vectors and empty AAV capsid particles were titrated using a quantitative real-time polymerase chain reaction (qPCR)[55,56]. The transgene expression cassettes for hF.IX, Gaa, and UGT1A1 were under the control of the apolipoprotein E (Apo E) enhancer/alpha1-antitrypsin (hAAT) liver-specific promoter. The expression cassettes of luciferase (luc), AAV8 capsid protein VP1, RFP, GFP, and SEAP were under the control of a CMV promoter. All vectors were free of endotoxin.

**Animal studies.** Animal studies were performed in accordance to the American and European legislations and approved by the intuitional ethical committee of the Centre d'Exploration et de Recherche Fonctionnelle Expérimentale (Evry, France, protocol number APAFIS 3055-20151019213299180) and by the animal care and use committee of Avastus Preclinical Services (Cambridge, Massachusetts). C57BL/6 mice were obtained from Jackson Laboratories. All mice were 4- to 8-weeks-old male at the onset of the experiments. Experimental procedures using non-haplotype matched male cynomolgus monkeys (Macaca fascicularis) were approved by the institutional animal care and use committee (protocol number APAF1S 2651- 2015110216583210 15) and were conducted at Nantes veterinary school (Oniris, Nantes, France). Animals were randomized to treatment groups at the beginning of each study and the investigator was blinded to the group allocation. For mouse studies, the minimal sample was determined[57] to allow to detect significant differences among treatment groups. In all mouse experiments, with the exception of Figure 5b (n = 4), a minimum of five animals per group was used.

**Efficacy studies.** Male C57BL/6 mice were treated intravenously (i.v.) with an AAV8 vector alone or together with SVP[Rapa] or SVP[empty], animals were then challenged, unless otherwise indicated, with an AAV8 vector alone or with SVP[Rapa] or SVP[empty]. Transgene, timing of injection and vector dose are indicated in figures legends. SVP[Rapa] was dosed at 50 µg when co-administered with

a vector dose of $4 \times 10^{11}$ vg kg$^{-1}$ and at 200 μg when co-administered with $4 \times 10^{12}$ vg kg$^{-1}$ AAV8.

**Tregs and B cell depletion**. For Tregs depletion, mice were treated with intraperitoneal injections of 200 μg of anti-CD25 antibody (PC61, BIO X Cell, West Lebanon, NH) or with PBS on days 19 and 20 after the initial injection of an AAV8-luc vector ($4 \times 10^{12}$ vg kg$^{-1}$) with SVP[Rapa] or SVP[empty]. On day 21, all animals received the intravenous injection of AAV8-hF.IX vector ($4 \times 10^{12}$ vg kg$^{-1}$) alone.

Mice were depleted of B cells by intraperitoneal injection of 250 μg of anti-CD20 antibody (Thermo Fisher Scientific, Waltham, MA). Two days after depletion, mice were treated with the intravenous injection of an AAV8-RFP vector ($4 \times 10^{11}$ vg kg$^{-1}$) alone or with SVP[Rapa].

**Adoptive transfer experiments**. For the memory T cell experiment, male C57BL/6 mice were left untreated or treated with a dose of $4 \times 10^{11}$ vg kg$^{-1}$ of an AAV8-RFP vector. 62 days later, spleens were collected and CD4$^{+}$ T cells were negatively selected (MACS, Miltenyi Biotech, Bergisch Gladbach, Germany). The enriched cells were confirmed to be >90% CD3$^{+}$CD4$^{+}$ T cells by flow cytometry (Supplementary Fig. 11) and adoptively transferred into recipient C57BL/6 mice at a dose of $10^{7}$ cells. Recipient mice were then treated with an AAV8-RFP vector ($4 \times 10^{11}$ vg kg$^{-1}$) together with SVP[Rapa] or SVP[empty] one day later. 21 days post adoptive transfer, mice were treated with AAV8-SEAP vector ($4 \times 10^{11}$ vg kg$^{-1}$) together with SVP[Rapa] or SVP[empty].

In an additional set of experiments, male C57BL/6 mice were left untreated or treated with AAV8-Luc vector ($4 \times 10^{12}$ vg kg$^{-1}$) together with SVP[Rapa] or SVP[empty]. 14 days later, spleens were collected and $3 \times 10^{7}$ splenocytes were adoptively transferred into recipient C57BL/6 mice. All animals were then treated with an AAV8-hF.IX vector ($4 \times 10^{12}$ vg kg$^{-1}$).

**Antigen-specificity studies**. Male C57BL/6 mice were treated i.v. with an AAV8-luc vector ($4 \times 10^{12}$ vg kg$^{-1}$) together with SVP[Rapa] or with SVP[empty], followed three weeks later by the i.v. injection of $4 \times 10^{12}$ vg kg$^{-1}$ of an AAV5-hF.IX vector or an AAV8-hF.IX vector. Additionally, mice were treated with same initial injection of AAV8-luc vector as described previously and challenged three weeks later with the subcutaneous injection of 100 μg of recombinant hF.IX protein (BeneFix®, Pfizer, Paris, France) formulated in complete Freund's adjuvant (Sigma-Aldrich, St. Louis, MO).

**Nonhuman primate experiments**. Before inclusion in the study, three male cynomolgus monkeys (*Macaca fascicularis*, 2 years old) were selected based on their lack of neutralizing antibodies to AAV8. At day 0, animals were randomized to the treatment groups and received an intravenous infusion (30 ml h$^{-1}$) of SVP[Rapa] (3 mg kg$^{-1}$ of rapamycin, $n = 2$, SVP[Rapa]#1 and SVP[Rapa]#2) or SVP[empty] ($n = 1$) immediately followed by the intravenous infusion of an AAV8-Gaa vector ($2 \times 10^{12}$ vg kg$^{-1}$)[58]. The investigator was blinded to the treatment group allocation of animals. One month later, each animal received a second infusion of SVP[Rapa] (3 mg kg$^{-1}$, $n = 2$, SVP[Rapa]#1 and SVP[Rapa]#2) or SVP[empty] ($n = 1$) followed by the infusion of an AAV8-hF.IX vector ($2 \times 10^{12}$ vg kg$^{-1}$).

Anesthesia performed before injection was induced first via intramuscular injection of demetomidine (30 mg kg$^{-1}$) and ketamine (7 mg kg$^{-1}$) then maintained with isoflurane (Vetflurane®) mixed with oxygen. AAV8 vectors and nanoparticles were infused in a volume of 10 or 5 ml, respectively, via the saphenous vein.

In all animal experiments, peripheral blood was collected and sera were isolated or immediately transferred to tubes containing citrates or EDTA to isolate plasma, at baseline and indicated time points. Spleen and inguinal lymph nodes were collected at necropsy in fresh RPMI medium and diverse organs were collected and stored at −80 °C for further analysis as indicated in figure legend. Cells counts in mouse peripheral blood were performed using MS9 automated cell counter with veterinary parameters and reagents (MS9; Schloessing Melet, Cergy Pontoise, France). For monkey samples, blood cell counts and biochemical analysis were performed at Nantes veterinary school (Oniris, Nantes, France).

**Anti-AAV antibody assays**. Antibody assays were performed using ELISA and in vitro neutralization assays[25,59]. For the ELISA assay, nunc maxisorp plates (Thermo Fisher Scientific, Waltham, MA) were coated with AAV particles ($2 \times 10^{10}$ particles per ml) and with serial dilution of purified immunoglobulin (IgG, IgG1, IgG2a, IgG2b, and IgG3 for murine samples; IgG and IgM for NHP samples) to generate a standard curve (Sigma-Aldrich, St. Louis, MO, and Fitzgerald Industries, Acton, MA, for mouse and monkey reagents, respectively). After overnight incubation at +4 °C, plates were blocked with PBS-0.05% Tween-20 containing 2% bovine serum albumin (BSA) and appropriately diluted samples were plated in duplicate. Samples were incubated for 3 h at room temperature. Plates were then washed, and the a secondary-antibody conjugated to HRP was added to the wells and incubated one hour at +37 °C (anti-mouse IgG-HRP, dilution 1:20,000, anti-mouse IgG1-HRP, dilution 1:10,000, anti-mouse IgG2a, dilution 1:5000, anti-mouse IgG2b, dilution 1:10,000, anti-mouse IgG3, dilution 1:10,000, all the secondary mouse antibodies conjugated to HRP were purchased

from Southern Biotech). For NHP samples, anti-monkey IgG-HRP (dilution 1:20,000, 43R-IG020HRP, Fitzgerald Industries, Acton, MA) and anti-IgM (dilution 1:5000, 2020-05, Southern Biotech) were used. Plates were then washed, and the presence of bound antibodies was detected by adding SIGMAFAST$^{TM}$ OPD substrate (Sigma-Aldrich, St. Louis, MO) following manufacturer's instructions and measuring OD at an absorbance of 492 nm.

The samples derived from experiments performed at Selecta Bioscience were analyzed with a different AAV8 antibody assay[60] and results are expressed as EC50 titers or top OD value. Briefly, 96-well plates were coated overnight with AAV8 vector at $2.2 \times 10^{9}$ vg ml$^{-1}$, washed and blocked with 1% casein the following day, then diluted serum samples (1:40) added to the plate and incubated overnight. Plates were then washed; a rabbit anti-mouse IgG specific-HRP (1:1500) was added to the wells and incubated for two hours. The presence of bound IgG antibodies was detected by adding TMB substrate and measuring at an absorbance of 450 nm with a reference wavelength of 570 nm.

Selected serum samples were also analyzed for anti-AAV neutralizing antibody titer using an in vitro cell-based test[24]. Briefly, in this assay, serial dilutions of heat-inactivated test samples were mixed with a vector expressing luciferase and incubated for one hour. After incubation, samples were added to cells and residual luciferase expression was measured after 24 h. The neutralizing titer was determined as the highest sample dilution at which at least 50% inhibition of luciferase expression was measured compared to a non-inhibition control. In this assay, a NAb titer of 1:10 represents the titer of a sample in which after a 10-fold dilution a residual luciferase signal lower than 50% of the non-inhibition control is observed.

**Factor IX plasma levels determination**. Plasma levels of human F.IX transgene were determined with an ELISA assay[29,59]. The detection of hF.IX antigen levels in mouse plasma was performed using the monoclonal antibodies against hF.IX (GAFIX-AP, Affinity Biologicals, Ancaster, Canada) for coating and anti-hFIX-HRP for detection (GAFIX-APHRP, Affinity Biologicals, Ancaster, Canada). In NHP samples, anti-hFIX antibody (MA1-43012, Thermo Fisher Scientific Waltham, MA), and anti-hF.IX-HRP antibody (CL20040APHP, Tebu-bio, Le Perray-en-Yvelines, France) were used for coating and detection, respectively. The presence of hF.IX antigen levels was determined by adding SIGMAFAST$^{TM}$ OPD substrate (Sigma-Aldrich) following manufacturer's instructions and measuring OD at an absorbance of 492 nm.

**AAV vector genome copy number determination**. Tissue DNA was extracted from whole organ using the Magna Pure 96 DNA and viral NA small volume kit (Roche Diagnostics, Indianapolis IN) according to the manufacturer's instructions. Vector genome copy number (VGCN) was performed using qPCR with primers specific for hAAT promoter for mice and for the hF.IX transgene cDNA for NHPs. Titin and albumin were used as normalizing genes in murine or NHP samples, respectively. VGCN was quantified by TaqMan real-time PCR with the ABI PRISM 7900 HT sequence detector (Thermo Fisher Scientific, Waltham, MA). The sequence of primers and probes is listed in Supplementary Table 4.

**Immunohistochemistry**. For Gaa, hF.IX and UGT1A1 staining, multiple wedge biopsies (left, right, quadrate and caudate lobes) from livers were fixed in 4% paraformaldehyde (PFA) for 10 min at room temperature (RT). Tissues were then incubated in 15% sucrose overnight then in 30% sucrose. The tissues were next placed in OCT, frozen in cold isopentane and subjected to cryosection (8 μm). Tissue sections were blocked with 5% BSA in PBS plus 4% goat serum and 0.1% Triton for 2 h at RT. Tissue Samples were incubated overnight at +4 °C with a monoclonal antibody against Gaa (clone EPR4716(2), dilution 1:100, Abcam, Cambridge, UK) and hF.IX (clone MA1-43012, dilution 1:100, Thermo Fisher Scientific Waltham, MA) or with monoclonal antibody against GFP (clone 1020, dilution 1:300, AVES Labs, Tigard, OR) and serum from rats immunized with UGT1A1 protein (Dilution 1:100). Following washes in PBS with 0.05% Tween-20, samples were incubated with secondary antibodies conjugated to Alexa Fluor 488, Alexa Fluor 555, or Alexa Fluor 594 (dilution 1:400, Thermo Fisher Scientific Waltham, MA). Following incubation and washes, sections were counterstained with Dapi. Images were captured using Confocal Microscope Leica SP8 (Leica MicroSystems, Wetzlar, Germany). Analysis of images was performed using the Cellprofiler software; staining signal was isolated from background signal (intensity 1 to 19 random units). A minimum of four fields per animal were acquired and analyzed to generate the data on distribution of signal intensity. The signal intensity was then divided by the surface in mm$^{2}$ to obtain number of positive hepatocytes per mm$^{2}$.

**Flow cytometry analysis**. Single-cell suspensions from spleen and lymph nodes were prepared and stained for the surface markers with different fluorochrome combinations: anti-CD3 (APC-eFluor780, clone 17A2, dilution 1:300, BD Biosciences, San Jose, CA), anti-CD4 (PB, clone RM4-5, dilution 1:200, BD Biosciences), anti-CD25 (PE, clone PC61, dilution 1:300 BD Biosciences), anti-CD45R (FITC, clone RA3.6B2, dilution 1:300, Thermo Fisher Scientific Waltham, MA), anti-CD19 (PE, clone 1D3, dilution 1:300 BD Biosciences), anti-IgD (APC-eFluor780, clone 11-26 c.2a, dilution 1:200, BD Biosciences), anti-GL7 (BV421,

clone GL7, dilution 1:100, BD Biosciences), anti-CD95 (PE-Cy7, clone Jo2, dilution 1:200, BD Biosciences), and anti-PD⁻1 (BV605, clone J43, dilution 1:200, BD Biosciences), followed by cell viability staining using Fixable Live/Dead kit (Biolegend, San Diego, CA) according to manufacturer's instructions. For the detection of mouse CXCR5, cells were first stained with biotin conjugated anti-CXCR5 or with biotin conjugated with isotype control (clone SPRCL5, dilution 1:50, Thermo Fisher Scientific) followed by streptavidin-PE (dilution 1:200, Thermo Fisher Scientific). Intracellular staining of FoxP3 (APC, clone FJK-16s, dilution 1:40, Thermo Fisher Scientific) was performed after fixation and permeabilization using murine FoxP3 buffer kit (Thermo Fisher Scientific) according to manufacturer's instructions. All antibodies were used at one test per $10^6$ cells. Samples were acquired using Sony Spectral cell analyzer SP6800 (Sony Biotechnology Inc., San Jose, CA). Data analysis was performed using the SP6800 software (Sony Biotechnology Inc. San Jose, CA) and FlowJo software (Tree Star, Ashland, OR).

**Anti-AAV T and B cell ELISpot assays.** Spleens were isolated, single-cell suspensions were then prepared and $2 \times 10^5$ cells were stimulated for 24 h with an AAV8 peptide pool[6] (1 μg ml⁻¹ of each peptide, Mimotopes, Victoria, Australia) in 100 μl of complete medium (RPMI supplemented with 10% FBS, Pen/Strep and L-Glutamine) (Thermo Fischer Scientific, Waltham, MA). Medium only was used as negative control and ConA (Concanavalin A Lectin Type IV, dilution 1:1000, Sigma-Aldrich, St. Louis, MO) was used as positive control. Additionally, cells were also stimulated in vitro for 7 days with AAV8 peptides.

For quantification of AAV8 antibodies secreting cells (ASC), 96-multiscreen plates (Merck-Millipore, Seattle, WA) were coated with empty AAV8 particles (1 × $10^{10}$ particles per well). After overnight incubation at +4 °C, plates were washed three times and blocked with complete medium (RPMI supplemented with 10% FBS, Pen/Strep and L-Glutamine) (Thermo Fischer Scientific) for 2 h at RT. $2 \times 10^5$ splenocytes per well and $5 \times 10^5$ splenocytes per well were incubated for 24 h at 37 °C and 5% $CO_2$. Plates were then washed and ASC were detected by incubation with biotin conjugated anti-IgG (for murine samples, Southern Biotech, Brimingham, AL; for NHP samples, Mabtech, Nacka Strand, Sweden) or biotin conjugated anti-mouse IgM (southern Biotech, Brimingham, AL) for 2 h at RT. After three washes, streptavidin-ALP (Mabtech, Nacka Strand, Sweden) was added for 1 h at RT, followed by the addition of the substrate for spot development (Mabtech). All antigens and controls were tested in triplicate. The plates were read on AID ELISpot reader system using the ELISpot 6.0 iSpot software (Autoimmun Diagnostika, Strassberg, Germany).

**RNA extraction and RT-PCR.** Total RNA were isolated from livers using the Magna Pure 96 cellular RNA large volume kit (Roche Diagnostics, Indianapolis IN) according to the manufacturer's instructions. RNA samples were then pretreated with DNAse (Thermo Fischer Scientific, Waltham, MA) and reverse transcribed using RevertAid H minus first strand cDNA synthesis kit (Thermo Fischer Scientific). RT-qPCR was performed using SybrGreen (Thermo Fisher Scientific) with primers specific for CD8 (Supplementary Table 4). Gene expression was calculated using the $\Delta\Delta C_t$ method relative to GAPDH housekeeping gene (Supplementary Table 4) and by normalizing to the mean expression of untreated mice.

**Statistical analysis.** Statistical analysis was performed using GraphPad Prism (GraphPad Software Inc, La Jolla, CA) version 6.0. Analysis of statistical significance between two groups was assessed using the two-tailed unpaired $t$-test. For multiple group comparisons, one-way or two-way ANOVA with Tukey multiple comparison test was used.

## Data availability
Data supporting the findings of this manuscript are available from the corresponding author upon reasonable request.

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

## Acknowledgements

This work was supported by Genethon and Selecta Bioscience, by the European Union, FP7-PEOPLE-2012-CIG Career Integration Grant, Grant Agreement No. 333628 (Nos-Mod to F.M.), ERC-2013-CoG Consolidator Grant, Grant Agreement No. 617432 (MoMAAV to F.M.), European Union's research and innovation program under Grant Agreements No. 667751 (Myocure, to F.M.), E-Rare2 grant SMART-HaemoCare (to F. M.), and the Bayer Early Career Investigator Award (to F.M.). A.Meliani was supported by a grant of the DIM-Biotherapies Ile-de-France.

## Author contributions

A.Meliani, F.B., S.M., A.Michaud, C.R., and P.I. performed most of the experiments and data analysis. C.L. contributed to the study in NHP. R.H., A.V., L.v.W., O.C., and G.M. contributed to the experimental activities. F.C., S.C., and M.S.S. produced the AAV vectors used in the studies. G.R. and H.C.V. contributed to the analysis data. A.Meliani, F.B., F.F., T.K.K., and F.M. designed the experiments, provided critical insights into the research activities and wrote the paper. F.M. supervised the project.

## Additional information

**Competing interest**F.M. consulted for Selecta Bioscience and Spark Therapeutics. F.M. received research support from Selecta Bioscience and is a current employee of Spark Therapeutics. C.R, A. Michaud, P.I., and T.K.K. are employees of Selecta Bioscience. The remaining authors declare no competing interests.

