## [Peer Review File · Nature Communications]

Reviewers' comments:

Reviewer #1 (Remarks to the Author):

This work is generally well done, timely, and has potentially great importance to the burgeoning field of gene therapy. It should ultimately be published following addressing the following points. Wisely, the authors do not claim immune tolerance via their platform, but this should be made more clear as well as making more prominent the data hidden in a supplementary figure (SFigure 1) which shows quite clearly that co-administration of SVP RAPA at the time of each antigen exposure is critical for inhibition of the anti-AAV humoral response. Moreover, the failure of Treg depletion to robustly restore the antibody response speaks to other mechanisms explaining the immune suppression. The other major issue I have regards the claim of antigen specificity. This approach is not antigen specific in the sense that whatever may be presented in close context to the Rapaparticles will incur immune suppression as well and other antigens introduced in a proximal timeframe to the rapaparticles may similarly be subject to immune suppression as well as any antigens present by happenstance. Other minor points include the readout of antibody responses only in OD rather than showing titers. Finally, as regards the Gaa/HfIX studies, the use of Gaa in this setting is not meaningful as it is an endogenous enzyme and likely the antibody used does not discriminate the endogenous NHP gaa from the vector mediated gaa whose origin is not revealed.

Reviewer #2 (Remarks to the Author):

The authors aim to show that co-administration of rapamycin particles with AAV can modulate immune responses, specifically neutralizing antibodies and T cell responses. This would allow for vector administration and long term expression of the transgene of interest without detriment. Though it is evident that a lot of work has gone on to produce the results shown in this paper, the results are in places unclear and lack suitable controls. In addition there does not seem to be any consistency in the amount of rapamycin used between the experiments with no explanation as to why in some experiments 50ug of rapamycin was used while in others 200ug was used. Furthermore a weakness in the paper is the number of NHP animals that were used (2 experimental, 1 control, a total of 3) and the conclusions that were drawn from these experiments despite the extremely small numbers. I have concerns also relative to the immunostainings that are not very clear and authors should improve figure 3 and suppl figure 3, maybe also adding some WBs. In terms of novelty, the system was already published showing immune-tolerance to the transgene product (as stated in Discussion).

Major comments

1. Page 5: the authors talk about VGCN being lower in SVP[Rapa] vs naïve animals in Fig 1e but it is unclear which are naïve. Do they refer to non SVP[Rapa] treated animals as naïve or truly naïve animals, i.e. non treated animals. This requires clarification.
2. Page 5/6: the authors state that timing of co-administration of SVP appears to be critical for the inhibition of the anti-AAV humoral response but only show results for the co-administration of SVP[Rapa] at the first injection (Supp Fig 1a.b). Do they have data showing co-administration of SVP[Rapa] at the time of second injection only and does this compliment their statement?
3. Page 6: Unfortunately it was difficult to follow the evaluation of the ability of SVP[Rapa] to enable a boost in transgene expression following a repeated dose as Fig 1g did not relate to the text nor did Fig 1h. Also there is no Fig 1i though this is mentioned in the text and nothing is related to Fig 1f. Please clarify the text and figures. Is it errors with the letters are present here: 1g is actually 1f, 1h is 1g and 1i is 1h.

4. Did the authors look at results from AAV injections without co-administration? Were the results similar to those with co-administration with SVP[Rapa]? Clarification is required as to whether the SVP[Rapa] is having an effect on the results.

5. This reviewer appreciates that NHP work is expensive but is doubtful as to whether conclusions can be drawn from 2 experimental animals and one control animal. A number of issues have arisen as a consequence.

a. Were the animals haplotype matched or at least closely haplotype matched?

b. The authors talk about SVP administration not appearing to have an impact on clinical chemistry or hematology parameters. Why is there no table for the control animal? In table 1 and 2 there is a huge increase in C reactive protein but this is not discussed. Also C reactive protein is misspelt in Table 2 as C relative protein. There also seems to be slightly elevated levels of leucocytes and neutrophils at certain time points in animal 1. Did the animal have an infection or was this a consequence of the protocol? This was not seen with the other experimental animal and needs to be discussed. Also as a unit of measure, there is a formatting issue for Reticulocytes to platelets which reads 103/l in Table 1 which I think should read 103/L. In fact all units pertaining to litres should be with a capital L.

c. Fig 2e and Fig 2g show statistical analysis significance for the data. This reviewer believes that the statistics used here are incorrect as these are technical replicates and not biological replicates. One way Anova with Bonferroni post hoc test is used to determine the significance between the means of different individual groups of data and not the mean of one individual data set. This needs to be corrected as it is misleading.

d. This reviewer does not believe the authors can state that 'these results demonstrate that inhibition of anti-AAV antibody formation mediated by SVP[Rapa] can be safely and effectively scaled up to a large animal model of gene transfer' with just 2 animals.

6. The authors headline a new results section with 'Repeated vector administration enhances targeting of different hepatocyte populations'. However the section only investigates hepatocytes in liver parenchyma. This is misleading and needs to be renamed.

7. The authors state that there were similar levels of Gaa in all three animals but the data in Fig 3a does not show this. If levels were similar the amount of Gaa should be similar in all three pictures but it is evident that there is much less Gaa in SVP[empty]. If the authors had shown the pictures for the different stainings individually before merging this would have been more evident. Please justify.

8. Why was 200ug of Rapamycin used for the mouse work to look at hepatocyte transduction and why was a different system used, ie why did the authors use GFP and UDP and not Gaa and hf.IX as for the NHP studies? Please justify. (Supp Fig 3)

9. The authors state 'These results indicate that AAV gene transfer with SVP[Rapa] allows for enhancement of liver transduction via targeting both cells already transduced and new populations of hepatocytes upon vector readministration' (Page 8). This sentence is confusing and this reviewer would like to know what 'both cells' refers to. The sentence needs to be rewritten to make the statement clearer.

10. Page 8: the authors evaluated T cell infiltrates by quantitative RT-PCR. Firstly, the presence of mRNA does not indicate the presence of the protein and thereby the presence of CD8 T cells. In addition, CD8 is also present on Dendritic cells, so how did the authors discern between CD8 T cells and CD8+ dendritic cells? This reviewer believes the statements 'CD8 T cell infiltrates were evident in control mice' and 'the level of CD8 T cells detectable in SVP[Rapa] treated animals was similar to that of naïve mice' are misleading as mRNA was quantified consequently should be rewritten and clarified.

11. Page 9: the authors state that 'germinal centre B cells were decreased in SVP[Rapa]-treated

animals (Fig 4e, f)' However it is unclear from the text and the figure legend to which group the FACs plot of Fig4e is referring to. Also why is the corresponding group not shown? This is pertinent to show that the gating strategy in both groups is identical.

12. Supp Fig 6: Why were female mice used for this experiment?

13. Can the authors please justify why they chose Day 80 for their memory T cell responses to AAV? It is well established that recall of T cell memory responses peaks around 48 hours after secondary injection.

14. The authors isolate CD4+ T cells by negative selection. As this incurs removal of all other populations of cells, do they have data to prove that they only have CD4+ T cells in the population that they transferred?

15. Fig 6b, c, and d: it is unclear which group the results are referring to. Which results correspond to the groups injected with AAV8-hf.IX, AAV5-hf.IX and hf.IX/CFA? This is unclear and needs addressing.

16. In addition to using a depleting antibody to CD25 (clone PC61) to determine the role of Tregs in immunomodulation, did the authors also consider using a blocking antibody (clone 7D4) to investigate the role of Tregs? Also did the authors look at the other populations that were depleted using PC61 such as B cells and NK cells? Perhaps their absence could have modulated the production of anti-AAV antibodies.

17. Fig 1: authors should clarify on the text that only three conditions are showed for the two consecutive administrations: 1st Rapa 2nd Rapa particles (blue), 1st empty 2nd empty particles (red) or 1st no particles 2nd no particles (black) but there are no mixed conditions. Only some of them (1st Rapa, 1st empty and 1st no particles +2nd no particles) are showed in suppl fig 1.

18. Are the cells stained in figure 3 and suppl fig 3 really hepatocytes? Is there any way to confirm that? Double/triple stainings?

19. Fig 3: it would be good if authors show some lower magnification fields (40x is really too high magnification to judge): in the panel "a" it does not seem that 12% of cells stain positive. Please use more representative pictures. Also the staining is not very clear as there is a lot of background: with 10-20% of positive cells a western blot should also be possible to confirm the proteins' expression? Suppl fig 3: same as for figure 3: pictures at very high magnification have been used; here the staining is a little better (though the background is still very high) but why the overlap of green/red does not appear yellow in this figure? Also the staining should be quantified in mouse as done for human and reported in Suppl fig 3.

Minor comments

1. Fig 1 legend; should (f) read 'AAV8-SEAP alone or mixed'rather than 'AAV8-SEAP alone or admixed'?

2. Fig 1 legend; you have put (n=x) for (g) please clarify the number of animals used

3. In some figure legends the authors state how much rapamycin is used, but in others this information is absent.

4. Page 3 line 3: Human clinical gene therapy clinical trials with AAV: revise as "Human clinical gene therapy trials with AAV"

5. Line 5: limitation: correct with "limitations"

6. Line 6: degenerative diseases treatment: correct with "degenerative diseases, treatment"

7. Page 13, lane 1: delete "and" and put a comma instead.

8. I think that in discussion there is no need to repeat "Fig 1", "Fig 2" etc...

Reviewer #3 (Remarks to the Author):

The authors utilize a rapamycin-incorporating PLGA nanoparticle technology (SVP[Rapa]), currently under clinical development, to limit immunity against adeno-associated viral vectors (AAV) which are being investigated for gene delivery. In mice and non-human primates, the results indicate the ability to largely prevent antibody production specific to AAV when the AAV vector is co-delivered with SVP[Rapa]. The reduction of anti-AAV antibody production was accompanied by increased vector genome copy number and plasma levels of the respective protein. It was shown that administration, and re-administration, of AAV vectors prevented antibody responses if co-administered with SVP[Rapa], but not if administered with SVP[empty] or without SVP. The SVP[Rapa]-treated animals contained more Foxp3+ T cells, and an in vivo depletion of Tregs (with anti-CD25) resulted in increased anti-AAV IgG suggesting a role for Tregs in the mechanism of tolerance, as has been previously proposed. The experiments were well-conceived and the results convincingly demonstrate tolerance induction toward AAV capsid proteins. However, most of the outputs related to SVP[Rapa]-mediated tolerance and its proposed mechanism have been previously demonstrated in several mouse and non-human primate models as well as in clinical trials (see references 18-23). The manuscript lacks novelty and does not represent a significant advancement to the field or provide a mechanistic understanding of the tolerance mechanisms.

Major Comments:

1. While this investigation utilizes a different animal model than has been previously published with the SVP[Rapa] platform, it does not differentiate itself from previous publications, including several animal (mouse and non-human primate) models and a clinical trial in humans (refs 18-23), by making significant advances to the understanding of the SVP platform.

2. Figure 6: The authors claim that SVP[Rapa] is not globally immunosuppressive by demonstrating that tolerance is not induced to AAV5 vectors and human F.IX protein (+CFA) administered 21 days after initial treatment with AAV8 + SVP[Rapa]. However, as designed, this data does not address the concern surrounding immunosuppressive drugs which is that exposure to disease relevant antigens concurrent with immunosuppressive treatment may lead to unintentional tolerance to pathogenic agents (or cancer from neoplastic cells). In these experiments, the authors introduce exogenous antigens 3 weeks following administration of SVP[Rapa] particles. As they have previously shown (Figure 2b of Kishimoto et al. Nature Nanotechnology 2016), tolerance could not be induced if SVP[Rapa] was administered 2 days prior to antigen delivery. Given their previous finding, it is not clear why the authors chose a 3 week time point instead of a shorter time point (less than 2 days) because this experiment does not satisfactorily address the safety concerns of general immunosuppression.

3. Figure 3: The authors claim that "SVP[Rapa] allows for enhancement of liver transduction via targeting both cells already transduced and new populations of hepatocytes upon vector readministration." This is an overstatement of the observations. First, the transduction of hepatocytes is not a "targeted" process. It appears to occur in a random fashion, and thus it is expected that there would be single positive hepatocytes for genes 1 or 2, and that, by chance, there would be hepatocytes that are double-positive for both genes. Since this is an expected result, it seems more appropriate for a supplementary figure. Overall, I believe that describing SVP[Rapa] as an enhancer of viral transduction is misleading. SVP[Rapa] was not shown to affect the mechanisms of viral transduction, but did prevent the formation of neutralizing antibodies that

may prevent transduction by AAV vectors.

4. There are no known diseases associated with AAV, however, tolerizing to AAV could have unknown consequences since AAVs do infect numerous people. There were no safety considerations addressed experimentally, nor even discussed in this manuscript.

Minor Comments:

1. There is an inconsistency in the amount of rapamycin administered (in Figures 4 and 6 200 µg were administered, in Figure 5, 50 µg were administered). It would be helpful if the authors could comment on the rationale for changing the dose.

2. The methods section does not provide sufficient details related to the particle preparation and characterization that would be required for scientists to reproduce this work. There is no description of the loading of rapamycin in the SVP[Rapa] platform or the dose of particles administered in the mouse studies.

3. In the caption for Figure 1, the description for panels d and e do not clearly describe the positive control of AAV8-hF.IX injected into naïve mice. I am concerned about my interpretation of the results.

4. Figure 1g caption, n value is written as "n=x" instead of actual number of individuals.

5. The in vitro neutralizing antibody assay (e.g. Fig 1c, Fig 2d) was not described well enough to describe what is being measured. Where IgG and IgM are clearly defined by fundamental immunology, one must know the assay described in reference 24 to understand the term defined as "neutralizing antibody" here. A brief explanation of the assay in the methods or caption would be clarifying.

6. The caption for Figure 2f reads, "Plasma hF.IX antigen levels were quantified by ELISA at the indicated time points following liver-directed administration of AAV8-hF.IX vector." What does liver-directed indicate?

7. Typographical errors:

o On page 8, "Quantification of hepatocytes IX from..."

o On page 10, "Next, we assessed the ability of SVP[Rapa] to control of primed..."

8. It is unclear what the authors are indicating by "different liver lobes" in Figure 3 caption: "Dual immunofluorescence staining of Gaa and hF.IX in different liver lobes from non-human primates treated"

9. The caption for Figure 4a says n=10, but it looks like only 5 are represented.

10. The text on page 9, and the caption for Figure 4, describes cells from the liver, spleen, and lymph nodes. In these descriptions, please clarify which organs are being referred to, or the sum of all organs, when describing the cell types measured. Also, both spleens and lymph nodes contain germinal centers, so please specify.

11. Page 10: The following text should specify that the described effect was observed specifically in SVP[Rapa], not all SVP-treated animals (which would indicate SVP[empty] as well, "No anti-AAV antibodies were observed in SVP-treated recipient mice."

12. Figure 6e shows SVP[empty] injection on day 1 after adoptive transfer, but this is not described in the caption or the results section. It is not clear which is accurate, the protocol in the figure, or the caption and the text.

13. Figure 1g shows that tolerance is broken when a subsequent dose of AAV8 is administered with SVP[empty] 93 days after the initial treatment with SVP[Rapa], but Figure 6a shows that tolerance is more durable when a subsequent dose is given at 21 days (although it looks like antibody production is starting to increase at day 41). This is an interesting finding and worth discussion or investigation into the mechanisms of tolerance maintenance.

Answers to Reviewers' comments:

Reviewer #1 (Remarks to the Author):

This work is generally well done, timely, and has potentially great importance to the burgeoning field of gene therapy. It should ultimately be published following addressing the following points.

ANSWER: We thank the Reviewer appreciating the quality of our work and for supporting the publication of our manuscript. We also very grateful for her/his help in improving the quality of our work.

Wisely, the authors do not claim immune tolerance via their platform, but this should be made more clear as well as making more prominent the data hidden in a supplementary figure (SFigure 1) which shows quite clearly that co-administration of SVP RAPA at the time of each antigen exposure is critical for inhibition of the anti-AAV humoral response. Moreover, the failure of Treg depletion to robustly restore the antibody response speaks to other mechanisms explaining the immune suppression.

ANSWER: We agree with the Reviewer. We in fact believe that it is preferable to avoid immune responses to the AAV capsid without inducing tolerance to a viral antigen. We also agree that while Tregs are needed to completely abrogate humoral immune responses to the AAV capsid, other mechanisms are likely to mediate the observed effect of SVP[Rapa].

We gave more prominence to Figure S1 by amending the corresponding results section, stating more clearly that co-administration of SVP[Rapa] at the time of each vector infusion is needed to block immune responses to the capsid.

We also added a sentence to the discussion section, which now reads as follows:

“In the setting of AAV liver gene transfer, administration of SVP[Rapa] concomitant to vector exposure was sufficient to inhibit anti-capsid immune responses. Co-administration of SVP[Rapa] with each dose of AAV vector appeared to be essential to completely block anti-AAV antibody responses, as repeated vector administration without SVP[Rapa] resulted in antibody formation even when SVP[Rapa] was co-administered with the first vector dose (**Supplementary Fig. 1**). Consistently, data generated indicate that the effect of SVP[Rapa] is selective for the co-administered antigen, as we showed that there was no effect on the antibody response to a different AAV serotype or to an unrelated protein antigen administered at a later time point (**Fig 6b**).”

We also clarify the point about regulatory T cells by amending the discussion section of the manuscript, which now reads as follows:

“The inability of anti-CD25 treatment to completely restore the anti-AAV antibody response may reflect incomplete deletion of CD25⁺ T cells in lymphoid organs or involvement of CD25-negative regulatory cells¹. Alternatively, it is plausible that other mechanisms are mediating the effect of SVP[Rapa] on capsid immunogenicity, including the immunosuppressive effect of rapamycin.”

The other major issue I have regards the claim of antigen specificity. This approach is not antigen specific in the sense that whatever may be presented in close context to the Rapa particles will incur immune suppression as well and other antigens introduced in a proximal timeframe to the rapa particles may similarly be subject to immune suppression as well as any antigens present by happenstance.

ANSWER: We understand the Reviewer's concern. This is an important point and we agree that potentially any antigen present in meaningful amounts in the spleen around the time of SVP[Rapa] administration could potentially be subjected to immune modulation. Thus, it is important to point out that the immunomodulatory effect would apply to antigens co-presented at the time of SVP[Rapa] co-administration. It is however important to point out that data in mice suggest that the temporal window of immune suppression mediated by SVP[Rapa] is limited. In published experiments ², see Figure 2b), administration of SVP[Rapa] is in fact unable to block antibody formation in response to antigens when administered more than one day before the antigen. In addition, there is a spatial window of immune suppression in that the SVP[Rapa] selectively accumulate in the spleen and liver following iv administration. Thus antigens that drain primarily to regional lymph nodes (e.g. pathogens that infect the lung) are not likely to co-localize with the antigen-presenting cells that take up the SVP[Rapa] in the spleen. Additionally, initial clinical data, albeit with small number of patients, show no evidence of a higher rate of infections in patients receiving multiple doses SVP[Rapa] in conjunction with pegsitticase over time for the treatment of refractory gout (ClinicalTrials.gov [NCT02959918](https://clinicaltrials.gov/ct2/show/study/NCT02959918) and ³).

We now amended the manuscript in several point to reflect these considerations. In particular, instead of defining the action of SVP[Rapa] as "antigen-specific", we now define it as "antigen-selective" throughout the manuscript, to better reflect that the effect is selective for antigens co-presented at the time of SVP[Rapa] administration. We amended the title, abstract and discussion sections of the manuscript to reflect these changes.

Other minor points include the readout of antibody responses only in OD rather than showing titers.

ANSWER: AAV antibody data is presented as a top OD in two instances – in Fig. 5D and in Supplementary Fig. 9c. Fig. 5d illustrates the seropositive status of splenocyte donors with the aim of demonstrating that the donor animal had pre-existing immunity to AAV8. The antibody kinetics in the recipient animals in this study is presented as EC50 titers (Fig. 5e). The aim of Supplementary Fig. 9c was to show that anti-CD20 antibody treatment did not inhibit the antibody response to AAV. While we understand the Reviewer's comment, we do not believe that the antibody titer or concentration data would change the conclusions from these two experiments. In all other experiments, antibody responses are presented as concentrations ($\mu\text{g/ml}$) or as EC50 titers.

We amended the Materials and Methods section to specify that the experiments in which the readout is an OD were performed in the laboratory at Selecta Bioscience:

"The samples derived from experiments performed at Selecta Bioscience were analyzed with a different AAV8 antibody assay ⁴ and results are expressed as EC50 titers or top OD value."

Finally, as regards the Gaa/HFIX studies, the use of Gaa in this setting is not meaningful as it is an endogenous enzyme and likely the antibody used does not discriminate the endogenous NHP gaa from the vector mediated gaa whose origin is not revealed.

ANSWER: This is a good point and we apologize to the reviewer for omitting these necessary controls. We now amended **Figure 3** and added the necessary staining controls as **Supplementary Figure 5**. We hope the new images and additional data will be more convincing.

Reviewer #2 (Remarks to the Author):

The authors aim to show that co-administration of rapamycin particles with AAV can modulate immune responses, specifically neutralizing antibodies and T cell responses. This would allow for vector administration and long term expression of the transgene of interest without detriment. Though it is evident that a lot of work has gone on to produce the results shown in this paper, the results are in places unclear and lack suitable controls.

ANSWER: We thank the Reviewer for helping us improve the quality of our work. We hope that the revised version of our manuscript addresses the initial concerns brought to our attention.

While we understand the concern raised about controls, we would like to point out that we included the appropriate controls in all experiments. This point has been clarified as outlined in the specific answers below.

The vast body of results presented here is clear in demonstrating, for the first time, that the SVP[Rapa] technology has the potential to change the entire landscape of gene therapy by allowing for AAV vector readministration. In particular, it shows:

- That one single administration of SVP[Rapa] at the time of AAV vector administration can completely block anti-capsid immune responses;
- That the immunomodulatory strategy based SVP[Rapa] nanoparticles is scalable to nonhuman primates;
- That the effect of SVP[Rapa] appears highly selective to the antigen presented at the time of administration.

Importantly, our work provides critical insights into the mechanism of action of the technology, which support the conclusion that the SVP[Rapa] technology is not simply an immunosuppressive therapy but in fact acts also via induction of Tregs.

In addition there does not seem to be any consistency in the amount of rapamycin used between the experiments with no explanation as to why in some experiments 50ug of rapamycin was used while in others 200ug was used.

ANSWER: This point was addressed below. We would like to point out that the use of different doses of SVP[Rapa] was consistent across experiments.

Higher vector doses are associated with greater immunogenicity. Hence the 50 µg dose of SVP[Rapa] was used with vector doses of 4E11 vg/kg while the 200 µg dose of SVP[Rapa] was used for vector doses of 4E12 vg/kg. We have clarified this in the Methods section.

Furthermore a weakness in the paper is the number of NHP animals that were used (2 experimental, 1 control, a total of 3) and the conclusions that were drawn from these experiments despite the extremely small numbers.

ANSWER: In this pilot study, we have tried to be mindful of the principles of the 3 Rs (replacement, reduction, and refinement) which has been adopted and advocated by animal welfare groups (AAALAC), academic and non-profit organizations, biopharmaceutical companies, regulatory agencies (FDA and EMA), and our local IACUC committee with respect to usage of nonhuman primates. The goal of the study was to evaluate if the general observation of anti-AAV antibody mitigation with SVP[Rapa] observed in mice could be translated to nonhuman primates. A single monkey was used as a control receiving AAV alone, as it is well established that AAV induces an immune response that prevents productive re-dosing of AAV. Two monkeys were used to study the effect of SVP[Rapa] on AAV immunogenicity and vector re-dosing. The data are reasonably unambiguous, and therefore provide evidence that suggest that SVP[Rapa] can enable re-dosing in nonhuman primates. A larger follow up study in NHP or a clinical study in humans will be required to determine how broadly applicable this finding is across a larger population.

We acknowledge the limitations of the current study and have changed the description of the study to make clear that it is a pilot study, softened the conclusion (from 'demonstrate' to 'suggest'), and state in the discussion that "Future studies in larger cohorts of nonhuman primates will be required for the clinical translation of the results shown here".

I have concerns also relative to the immunostainings that are not very clear and authors should improve figure 3 and suppl figure 3, maybe also adding some WBs.

ANSWER: We now clarify this point and added new images of the liver immunostaining and relative controls.

In terms of novelty, the system was already published showing immune-tolerance to the transgene product (as stated in Discussion).

ANSWER: SVP[Rapa] have been demonstrated to prevent the formation of anti-drug antibodies to Factor VIII (Factor IX was not tested) protein as well as other protein therapeutics. However AAV vectors arguably present a greater challenge in that 1) as a 25 nm viral particle displaying repetitive antigen epitopes it is considerably more immunogenic than typical protein therapeutics and 2) the degree of antibody mitigation has to be virtually complete, as even low levels of anti-AAV antibodies are capable of completely blocking AAV transduction. More importantly, the novelty of the current findings is that SVP[Rapa] enables successful re-administration of AAV vectors. The ability of readminister AAV vectors is the Holy Grail of gene therapy. To date no study has been successful in demonstrating that it is possible to readminister AAV vectors in mice and in nonhuman primates by means of a simple intervention (e.g. ⁵⁻⁷). We understand that the mechanism of action of this technology may not be completely new, but the use of SVP[Rapa] in the context of AAV gene transfer is definitely novel and has stirred a lot of excitement in the field of gene therapy. Of note, the ability of readministering AAV unlocks the potential of the technology in the treatment of diseases with early lethality, in which the effect of gene transfer is not expected to last long-term. It also potentially allows the treatment of neuromuscular

disorders in which systemic targeting of the entire body may need to be achieved via repeated vector administrations.

We clarified these points in the discussion section of the manuscript, which now reads as follows:

“These findings are of high relevance to gene therapies for diseases with early lethality⁸, in which early intervention may require subsequent treatments to maintain efficacy over time⁹. They also address the need for strategies to carefully dose-titrate AAV vectors to avoid overexpression¹⁰, or to target systemic diseases¹¹ in which multiple vector administrations are likely needed to achieve full therapeutic efficacy¹².

Despite numerous attempts, the ability to readminister AAV vector still represents an unmet goal for the field of in vivo gene transfer. Some degree of success has been achieved in the context of intramuscular gene transfer¹³ or gene transfer to immunoprivileged body sites^{14,15}, in which vector neutralization by circulating antibodies is not a concern¹⁶.”

We also discussed more extensively other possible approaches to vector readministration, to better support the fact that our work is novel and highly promising and that it represents a potentially simple and highly effective solution to a complex important issue for the entire field of gene transfer.

“Plasmapheresis has also been proposed to readminister AAV vectors¹⁷, however this maneuver has several limitations, as human studies indicate that repeated sessions will be needed to eliminate anti-AAV antibodies present in the extravascular space, and that this approach is unlikely to completely eradicate high-titer antibodies¹⁸. Similarly, capsid switching has been proposed¹⁹, however this is also a very complex procedure from a drug development and manufacturing perspective, and it is further complicated by high degree of cross-reactivity of anti-AAV antibodies²⁰.”

Major comments

1. Page 5: the authors talk about VGCN being lower in SVP[Rapa] vs naïve animals in Fig 1e but it is unclear which are naïve. Do they refer to non SVP[Rapa] treated animals as naïve or truly naïve animals, i.e. non treated animals. This requires clarification.

ANSWER: We apologize to the Reviewer for the lack of clarity. We now specified that these animals were naïve and uniquely dosed with AAV8-hF.IX vector at day 21 to provide a control of the efficiency of liver transduction and anti-capsid antibody formation in naïve animals. We now clarified this point both in the figure 1 legend and in the results section which now reads as follows:

“Accordingly, successful vector readministration, measured by plasma levels of hF.IX transgene product, was achieved only in animals receiving SVP[Rapa] and in a control group of naïve animals dosed at day 21 with AAV8-hF.IX vector only (**Fig. 1d**).”

2. Page 5/6: the authors state that timing of co-administration of SVP appears to be critical for the inhibition of the anti-AAV humoral response but only show results for the co-administration of SVP[Rapa] at the first injection (Supp Fig 1a.b). Do they have data

showing co-administration of SVP[Rapa] at the time of second injection only and does this compliment their statement?

ANSWER: We understand the Reviewer's concern. Figure 1d shows that administration of an AAV8-Luc vector with SVP[empty] completely blocks transduction of an AAV-hF.IX vector given at day 21 (in blue). Thus, a vector given with SVP[Rapa] at the time of the second administration only would not result in any meaningful transduction of the liver.

This control was indeed originally included in the experimental design. As shown in **Supplementary Fig. 2**, administration of AAV-Luc + SVP[empty] at day 0 and AAV-hF.IX + SVP[Rapa] at day 21 results in neutralization of the AAV-hF.IX vector (as shown by the lack of hF.IX transgene expression).

3. Page 6: Unfortunately it was difficult to follow the evaluation of the ability of SVP[Rapa] to enable a boost in transgene expression following a repeated dose as Fig 1g did not relate to the text nor did Fig 1h. Also there is no Fig 1i though this is mentioned in the text and nothing is related to Fig 1f. Please clarify the text and figures. Is it errors with the letters are present here: 1g is actually 1f, 1h is 1g and 1i is 1h.

ANSWER: We apologize for the mislabeling of the figure panels. We now corrected the text describing the panels, we hope this will help following the experimental design and results.

4. Did the authors look at results from AAV injections without co-administration? Were the results similar to those with co-administration with SVP[Rapa]? Clarification is required as to whether the SVP[Rapa] is having an effect on the results.

ANSWER: We understand the Reviewer's concern. We would like to reassure the Reviewer that every experiment was performed multiple times and with the proper controls. Specifically, to clearly show the effect of SVP[Rapa] in each experiment shown a control group of animals treated with SVP[empty], nanoparticles not containing rapamycin, was included. As shown, all animals receiving vector with the control excipient developed anti-AAV antibodies which made vector re-administration ineffective (see in particular Figures 1 and 2). We amended the figure legends in the manuscript to specify that SVP[empty] animals represent the control group in each experiment.

5. This reviewer appreciates that NHP work is expensive but is doubtful as to whether conclusions can be drawn from 2 experimental animals and one control animal. A number of issues have arisen as a consequence.

ANSWER: This point was addressed above. Once more we would like to point out that the results obtained are quite clear-cut and informative and multiple readouts of the data are shown and are highly consistent (expression levels, neutralizing antibodies, IgG and IgM levels, gene copy number in the liver). Furthermore, the results in non-human primates are consistent and supported by the data obtained in mice in the several experiments shown. As discussed, we amended the text of the manuscript to reflect the fact that this was a pilot study and larger studies will be needed to translate the approach to humans.

a. Were the animals haplotype matched or at least closely haplotype matched?

ANSWER: The animals were purposely not haplotype matched. Using haplotype matched animals is something rather complicated to do and unusual for this kind of experiments. The

study shown here is similar to other published gene therapy studies^{5-7,21}, in which no haplotype matched animals were ever used. We would like to point out that non-human primates are usually used to confirm that a given approach is scalable from small animal models (e.g. mice) to a model closer to humans. This also includes the inherent variability deriving from outbred species, more closely resembling the clinical reality. We specified the fact that animals were not haplotype matched in the material and methods section.

b. The authors talk about SVP administration not appearing to have an impact on clinical chemistry or hematology parameters. Why is there no table for the control animal? In table 1 and 2 there is a huge increase in C reactive protein but this is not discussed. Also C reactive protein is misspelt in Table 2 as C relative protein. There also seems to be slightly elevated levels of leucocytes and neutrophils at certain time points in animal 1. Did the animal have an infection or was this a consequence of the protocol? This was not seen with the other experimental animal and needs to be discussed. Also as a unit of measure, there is a formatting issue for Reticulocytes to platelets which reads 103/I in Table 1 which I think should read 103/L. In fact all units pertaining to litres should be with a capital L.

ANSWER: We apologize for the omission. We now included the table with the data from the control animal. We also corrected the tables to reflect the Reviewer's suggestions.

We did comment on the increase in C reactive protein in the SVP[Rapa] animals. The sentence in the results section reads as follows:

"There was no alteration of clinical chemistry or hematology parameters in the SVP[Rapa] animals except for a transient increase in C reactive protein around the time of treatment (Supplementary Tables 1-3)."

c. Fig 2e and Fig 2g show statistical analysis significance for the data. This reviewer believes that the statistics used here are incorrect as these are technical replicates and not biological replicates. One way Anova with Bonferroni post hoc test is used to determine the significance between the means of different individual groups of data and not the mean of one individual data set. This needs to be corrected as it is misleading.

ANSWER: While it is true that Figure 2e represents replicates of the same sample, Figure 2g represent measurements of vector genome copy number in samples collected from different liver lobes (left, right, caudate and quadrate). We do understand the Reviewer's concerns and, accordingly, we eliminated the statistical analysis from Figure 2e and instead indicated a threshold of 50 spot forming units per million cells, which is commonly used as a cutoff for positivity in this assay. We also modified the statistical analysis for Figure 2e and compared individual groups as suggested. The results in terms of significance are unchanged.

d. This reviewer does not believe the authors can state that 'these results demonstrate that inhibition of anti-AAV antibody formation mediated by SVP[Rapa] can be safely and effectively scaled up to a large animal model of gene transfer' with just 2 animals.

ANSWER: We agree that translating the results to humans will require further studies. To capture this point we amended the results section, last sentence, by stating that the results "suggest" that the approach is scalable to a large animal model of gene transfer:

“These results suggest that inhibition of anti-AAV antibody formation mediated by SVP[Rapa] can be safely and effectively scaled up to a large animal model of gene transfer.”

We also amended the discussion section by adding the following sentence:

“Future studies in larger cohorts of nonhuman primates will be required for the clinical translation of the results shown here.”

6. The authors headline a new results section with ‘Repeated vector administration enhances targeting of different hepatocyte populations’. However the section only investigates hepatocytes in liver parenchyma. This is misleading and needs to be renamed.

ANSWER: We thank the reviewer for pointing this out. We have changed to section title to: “Repeated vector administration increases the number of transduced hepatocytes in the liver parenchyma”

7. The authors state that there were similar levels of Gaa in all three animals but the data in Fig 3a does not show this. If levels were similar the amount of Gaa should be similar in all three pictures but it is evident that there is much less Gaa in SVP[empty]. If the authors had shown the pictures for the different stainings individually before merging this would have been more evident. Please justify.

ANSWER: We agree with the Reviewer that the data presented were not entirely clear. We would like to point out that we do not expect the levels of AAV-Gaa vector transduction to be significantly different across the three animals as this was the first vector administered at day 0.

To better represent the GAA staining in liver, we modified Figure 3 and added an additional supplementary figure:

Figure 3. Individual staining of Gaa and hF.IX are shown as well as merged images at high magnification.

Supplementary Figure 5. Individual staining of Gaa and hF.IX as well as merged images at various magnification levels. Additionally, we showed the negative control for these stainings (double staining in uninjected NHP).

Based on these new data, we conclude that the staining for Gaa is similar in all animals.

8. Why was 200ug of Rapamycin used for the mouse work to look at hepatocyte transduction and why was a different system used, ie why did the authors use GFP and UDP and not Gaa and hf.IX as for the NHP studies? Please justify. (Supp Fig 3)

ANSWER: We used the same SVP[Rapa] dose (200µg/mouse) in several of our experiments (Figure 1a-e; Figure 4; Figure 6) which resulted in good control of anti-AAV antibody responses in mice administered a vector dose of 4×10^{12} vg/kg. Therefore this dose was used also in the mouse experiment where we infused GFP and UGT1A1 expressing vectors.

As for the choice of the transgenes, the experiment in NHPs used secretable transgenes that we could track in plasma. The mouse study used non-secretable transgenes that are easy to detect in hepatocytes. It should be noted that GFP is highly immunogenic in nonhuman primates²², which does not allow to track transgene expression long-term. Since we also never tested human UGT1A1 expression in NHP liver, we preferred using GAA and hF.IX.

We clarified this point in the results section by adding the following sentences:

“Two vectors encoding for secreted transgenes were administered to the animals, one at day 0 encoding for alpha-acid glucosidase (AAV8-Gaa) and one at day 30 encoding for hF.IX (AAV8-hF.IX.). Both vectors were given at a dose of 2×10^{12} vg/kg.”

“Similar experiments were conducted in mice with two vectors encoding for the non-secreted transgenes green fluorescent protein (AAV8-GFP) and human uridine diphosphate (UDP) Glucuronosyltransferase 1A1 (AAV8-hUGT1A1)²³, administered at day 0 and 21, respectively (Supplementary Fig. 5a).”

9. The authors state ‘These results indicate that AAV gene transfer with SVP[Rapa] allows for enhancement of liver transduction via targeting both cells already transduced and new populations of hepatocytes upon vector readministration’ (Page 8). This sentence is confusing and this reviewer would like to know what ‘both cells’ refers to. The sentence needs to be rewritten to make the statement clearer.

ANSWER: We apologize to the Reviewer for the confusion. Our intent was to emphasize that the second vector administration enabled by SVP[Rapa] treatment doesn’t just transduce the same population of hepatocytes that were transduced with the first vector administration, it also allows for transduction of additional hepatocytes that were not previously transduced (i.e. cells that do not express the transgene encoded by the first vector).

We have revised the text to read: “These results indicate that repeated AAV vector administration enabled by SVP[Rapa] treatment can enhance liver gene transfer in part by increasing the number of hepatocytes that are transduced.”

10. Page 8: the authors evaluated T cell infiltrates by quantitative RT-PCR. Firstly, the presence of mRNA does not indicate the presence of the protein and thereby the presence of CD8 T cells. In addition, CD8 is also present on Dendritic cells, so how did the authors discern between CD8 T cells and CD8+ dendritic cells? This reviewer believes the statements ‘CD8 T cell infiltrates were evident in control mice’ and ‘the level of CD8 T cells detectable in SVP[Rapa] treated animals was similar to that of naïve mice’ are misleading as mRNA was quantified consequently should be rewritten and clarified.

ANSWER: We agree with the Reviewer and have revised the text to read: “We evaluated the liver of mice treated with AAV vectors and SVP[Rapa] or SVP[empty] for the presence of cells expressing CD8 mRNA by quantitative RT-PCR. CD8 mRNA expression was significantly elevated in mice treated with AAV vector and SVP[empty] control particles compared to naïve mice that did not receive AAV vector. In contrast, the level of CD8 mRNA detectable in animals treated with AAV vector and SVP[Rapa] was not significantly different from that of naïve mice.”

11. Page 9: the authors state that ‘germinal centre B cells were decreased in SVP[Rapa]-treated animals (Fig 4e, f)’ However it is unclear from the text and the figure legend to which group the FACS plot of Fig4e is referring to. Also why is the corresponding group not shown? This is pertinent to show that the gating strategy in both groups is identical.

ANSWER: The flow plot in figure 4e represents an example of the raw data for the staining of germinal center B cells. The specific sample was a mouse from the SVP[empty] group. The histogram in figure 4f is the representation of the combined data (frequency of CD95+GL-7+ cells) in the SVP[empty] vs. the SVP[Rapa] experimental groups.

We now clarified this point in the figure 4 legend.

12. Supp Fig 6: Why were female mice used for this experiment?

ANSWER: For long-term studies requiring multiple blood sampling time points, we have found that the attrition of mice due to fighting is much lower with female mice than with male mice. We also believe that it was important to show that the effects of SVP[Rapa] were not sex-specific. We do not believe that the sex of the animals had any influence on the outcome of the experiment.

13. Can the authors please justify why they chose Day 80 for their memory T cell responses to AAV? It is well established that recall of T cell memory responses peaks around 48 hours after secondary injection.

ANSWER: We agree with the Reviewer and we apologize for the misunderstanding. Animals were primed, then boosted 77 days after (when presumably a population of memory T cells would be established). On day 80, 72 hours after the second injection, the animals were sacrificed for ELISpot analysis. This is within the typical 48-72h optimal window for measuring memory T cell recall responses.

We now clarified this point in the manuscript:

Figure legend: "Spleens were collected at day 19 and day 80 after priming and boosting, respectively."

Results section: "T cell reactivity to the AAV capsid as measured by IFN- γ ELISpot at day 19 (after a single AAV dose) or day 80 (3 days after a repeat dose of AAV) was significantly lower in animals treated with SVP[Rapa] versus SVP[empty]."

14. The authors isolate CD4+ T cells by negative selection. As this incurs removal of all other populations of cells, do they have data to prove that they only have CD4+ T cells in the population that they transferred?

ANSWER: CD4-positive T cells were isolated by depletion of non-CD4 cells using a cocktail of biotin-conjugated antibodies against CD8a, CD11b, CD11c, CD19, CD45R (B220), CD49b (DX5), CD105, anti-MHC-class II, Ter-119 and TCR γ/δ as primary labeling reagent (Miltenyi Biotec Inc., Auburn, CA). The non-target cells were magnetically labeled with anti-biotin MicroBeads and then depleted from suspension by retaining them on a MACS Column in the magnetic field of a MACS Separator. The purity of the CD4 T cells was 93% as determined by flow cytometry (**Supplementary Fig. 11**).

We have added a statement to the Methods that reads: "The enriched cells were confirmed to be >90% CD3+CD4+ T cells by flow cytometry (**Supplementary Fig. 11**)."

15. Fig 6b, c, and d: it is unclear which group the results are referring to. Which results correspond to the groups injected with AAV8-hf.IX, AAV5-hf.IX and hf.IX/CFA? This is unclear and needs addressing.

ANSWER: We apologize for the lack of clarity. To help the reader follow the results, we modified the Figure 3 and the relative legend to clearly identify the experimental groups. We also more clearly referred to the figure's panels in the results section.

16. In addition to using a depleting antibody to CD25 (clone PC61) to determine the role of Tregs in immunomodulation, did the authors also consider using a blocking antibody

(clone 7D4) to investigate the role of Tregs? Also did the authors look at the other populations that were depleted using PC61 such as B cells and NK cells? Perhaps their absence could have modulated the production of anti-AAV antibodies.

ANSWER: We understand the Reviewer's concern. To confirm that some active tolerance mechanism was involved in the modulation of AAV immunogenicity mediated by AAV, we used two different experimental settings, 1) the adoptive transfer 2) Treg depletion. As stated above in response to Reviewer 1 comments, we believe our results are suggestive of a role of Tregs in the modulation of anti-AAV immune responses. However, additional mechanisms seem to be involved. Future studies will address this point in detail. Please see answers to Reviewer 1 for a more detailed discussion about Tregs.

Given that anti-AAV antibody levels in the control group receiving PC61 + AAV vector + SVP[empty] particles are statistically identical to the AAV vector + SVP[empty] control, it is unlikely that the PC61 administration influenced the outcome of the experiment.

We captured this last consideration in the corresponding results section:

"Conversely, PC61 administration together with SVP[empty] control had no effect on anti-AAV antibody levels."

17. Fig 1: authors should clarify on the text that only three conditions are showed for the two consecutive administrations: 1st Rapa 2nd Rapa particles (blue), 1st empty 2nd empty particles (red) or 1st no particles 2nd no particles (black) but there are no mixed conditions. Only some of them (1st Rapa, 1st empty and 1st no particles +2nd no particles) are showed in suppl fig 1.

ANSWER: We now clarified this point in the results section of the manuscript. As shown in the response to the Reviewer's question #2, we did test all the combinations of SVP[Rapa]/SVP[empty] regimens (SVP[Rapa]/ SVP[Rapa]; SVP[empty]/SVP[empty]; SVP[Rapa]/SVP[empty]; SVP[empty]/SVP[Rapa]; no SVP).

We specified the experimental conditions shown in Figure 1 by including the following sentence in the results section of the manuscript:

"Three experimental conditions were tested, i) administration of both vectors with SVP[Rapa], ii) administration of both vectors with SVP[empty] control, iii) administration of the AAV8-hF.IX vector at day 21 only (no additional treatment control)."

18. Are the cells stained in figure 3 and suppl fig 3 really hepatocytes? Is there any way to confirm that? Double/triple stainings?

ANSWER: We understand the Reviewer's concern. Given that the promoter driving the expression of the transgenes in the vectors used in the study is strictly hepatocyte-specific, we believe that the vast majority of cells positive for transgene expression are indeed hepatocytes.

19. Fig 3: it would be good if authors show some lower magnification fields (40x is really too high magnification to judge): in the panel "a" it does not seem that 12% of cells stain positive. Please use more representative pictures. Also the staining is not very clear as there is a lot of background: with 10-20% of positive cells a western blot should also be possible to confirm the proteins' expression? Suppl fig 3: same as for figure 3: pictures at very high magnification have been used; here the staining is a little better (though the background is still very high) but why the overlap of green/red does not appear yellow in

this figure? Also the staining should be quantified in mouse as done for human and reported in Suppl fig 3.

ANSWER: We now added additional images at lower magnification (new Supplementary Figure 5), which give a better idea of the extent of liver transduction in the different animals. Being both Gaa and hF.IX secretable transgenes expressed under the control of a liver specific promoter, it is not necessary to perform a Western blot to detect protein production. It is in fact evident that all animals have Gaa expression in plasma (see figure below). As this results was part of another publication (Puzzo et al., Science Translational Medicine 2017, Figure 7A)²⁴, Gaa plasma levels were not included in this manuscript to avoid results duplication. hF.IX levels are shown in **Figure 2F** in the current manuscript.

We also modified Supplementary Fig. 3 (now **Supplementary Fig. 6**) to include the information requested by the Reviewer.

Minor comments

1. Fig 1 legend; should (f) read ‘AAV8-SEAP alone or mixed’rather than ‘AAV8-SEAP alone or admixed’?

ANSWER: Corrected.

2. Fig 1 legend; you have put (n=x) for (g) please clarify the number of animals used

ANSWER: Corrected.

3. In some figure legends the authors state how much rapamycin is used, but in others this information is absent.

ANSWER: We apologize for the omission, we now stated the dose of rapamycin used in all figure legends.

4. Page 3 line 3: Human clinical gene therapy clinical trials with AAV: revise as “Human clinical gene therapy trials with AAV”

ANSWER: Corrected.

5. Line 5: limitation: correct with “limitations”

ANSWER: Corrected.

6. Line 6: degenerative diseases treatment: correct with “degenerative diseases, treatment”

ANSWER: Corrected.

7. Page 13, lane 1: delete “and” and put a comma instead.

ANSWER: Corrected.

8. I think that in discussion there is no need to repeat “Fig 1”, “Fig 2” etc...

ANSWER: While we understand that this may seem somewhat tedious, however we feel it is good to refer at least to some of the results in the discussion to help the reader navigate the large body of data described.

Reviewer #3 (Remarks to the Author):

The authors utilize a rapamycin-incorporating PLGA nanoparticle technology (SVP[Rapa]), currently under clinical development, to limit immunity against adeno-associated viral vectors (AAV) which are being investigated for gene delivery. In mice and non-human primates, the results indicate the ability to largely prevent antibody production specific to AAV when the AAV vector is co-delivered with SVP[Rapa]. The reduction of anti-AAV antibody production was accompanied by increased vector genome copy number and plasma levels of the respective protein. It was shown that administration, and re-administration, of AAV vectors prevented antibody responses if co-administered with SVP[Rapa], but not if administered with SVP[empty] or without SVP. The SVP[Rapa]-treated animals contained more Foxp3+ T cells, and an in vivo depletion of Tregs (with anti-CD25) resulted in increased anti-AAV IgG suggesting a role for Tregs in the mechanism of tolerance, as has been previously proposed.

The experiments were well-conceived and the results convincingly demonstrate tolerance induction toward AAV capsid proteins. However, most of the outputs related to SVP[Rapa]-mediated tolerance and its proposed mechanism have been previously demonstrated in several mouse and non-human primate models as well as in clinical trials (see references 18-23). The manuscript lacks novelty and does not represent a significant advancement to the field or provide a mechanistic understanding of the tolerance mechanisms.

ANSWER: We thank the Reviewer for appreciating the quality of our work and for helping us improving the presentation and discussion of the results reported. We hope the revised version of our manuscript will more convincingly support the novelty of our findings.

Major Comments:

1. While this investigation utilizes a different animal model than has been previously published with the SVP[Rapa] platform, it does not differentiate itself from previous publications, including several animal (mouse and non-human primate) models and a clinical trial in humans (refs 18-23), by making significant advances to the understanding of the SVP platform.

ANSWER: As stated above, in answer to reviewer 1, SVP[Rapa] has been demonstrated to prevent the formation of anti-drug antibodies to protein therapeutics. AAV vectors arguably present a greater challenge in that 1) it is a 25 nm viral particle displaying repetitive antigen epitopes and as a result is considerably more immunogenic than typical protein therapeutics and 2) the degree of antibody mitigation has to be virtually complete, as even low levels of anti-AAV antibodies are capable of completely blocking AAV transduction. More importantly, the novelty of the current findings is that the SVP[Rapa] enables successful re-administration of AAV vectors. The inability to re-dose AAV is considered to be a major challenge for the use of systemic gene therapy in pediatric patients.

These points have been clarified in the discussion section of the manuscript, which now reads:

“Despite numerous attempts, the ability to readminister AAV vector still represents an unmet goal for the field of in vivo gene transfer. Some degree of success has been achieved in the context of intramuscular gene transfer¹³ or gene transfer to immunoprivileged body sites^{14,15}, in which vector neutralization by circulating antibodies is not a concern¹⁶.”

We also discussed more extensively other possible approaches to vector readministration, to better support the fact that our work is novel and highly promising and that it represents a potentially simple and highly effective solution to a complex important issue for the entire field of gene transfer.

“Plasmapheresis has also been proposed to readminister AAV vectors¹⁷, however this maneuver has several limitations, as human studies indicate that repeated sessions will be needed to eliminate anti-AAV antibodies present in the extravascular space, and that this approach is unlikely to completely eradicate high-titer antibodies¹⁸. Similarly, capsid switching has been proposed¹⁹, however this is also a very complex procedure from a drug development and manufacturing perspective, and it is further complicated by high degree of cross-reactivity of anti-AAV antibodies²⁰.”

2. Figure 6: The authors claim that SVP[Rapa] is not globally immunosuppressive by demonstrating that tolerance is not induced to AAV5 vectors and human F.IX protein (+CFA) administered 21 days after initial treatment with AAV8 + SVP[Rapa]. However, as designed, this data does not address the concern surrounding immunosuppressive drugs which is that exposure to disease relevant antigens concurrent with immunosuppressive treatment may lead to unintentional tolerance to pathogenic agents (or cancer from neoplastic cells). In these experiments, the authors introduce exogenous antigens 3 weeks following administration of SVP[Rapa] particles. As they have previously shown (Figure 2b of Kishimoto et al. Nature Nanotechnology 2016), tolerance could not be induced if SVP[Rapa] was administered 2 days prior to antigen delivery. Given their previous finding, it is not clear why the authors chose a 3 week time point instead of a shorter time point (less than 2 days) because this experiment does not satisfactorily address the safety concerns of general immunosuppression.

ANSWER: We understand the Reviewer’s concern. Please see our response to Reviewer 1 regarding the temporal and spatial window of tolerance induced by SVP[Rapa]. We agree that this does not fully address questions about specificity or safety. We have softened claims about specificity by changing antigen-specific to antigen-selective and have added to the Discussion a statement that reads “Further evaluation of safety will have to be addressed in preclinical studies and clinical trials”. While large clinical trials are not feasible for gene therapy of rare inheritable diseases, Selecta plans to start a Phase 3 trial later this year of SVP[Rapa] in combination with a pegylated uricase enzyme in patients with severe gout, which should provide additional insight into the safety of SVP[Rapa]. Importantly, results obtained to date in the ongoing Phase I/II trials support the safety of SVP[Rapa] in humans.

In our experimental setting, we chose a 3-week interval to demonstrate selectivity to be consistent with the interval used in other experiments (e.g. Figure 1) testing vector re-administration.

We did conduct an experiment testing SVP[Rapa] administered one day prior to vector administration (please see **Supplementary Fig. 3**). The left panel shows anti-AAV8 antibody responses in mice (n=5) treated with AAV8-luc and SVP[Rapa] on the same day (Day 0) and then boosted with AAV8-hF.IX without SVP[Rapa] on day 14, while the right panel shows the antibody response in animals treated with SVP[Rapa] one day prior to administration of AAV8-luc followed by a boost injection of AAV8 alone on day 14. In this experiment, SVP[Rapa] co-administered with the first dose of AAV8 was more effective than SVP[Rapa] administered one day prior to the first dose of AAV8 (red squares in the two graphs in **Supplementary Fig. 3**).

3. Figure 3: The authors claim that “SVP[Rapa] allows for enhancement of liver transduction via targeting both cells already transduced and new populations of hepatocytes upon vector readministration.” This is an overstatement of the observations. First, the transduction of hepatocytes is not a “targeted” process. It appears to occur in a random fashion, and thus it is expected that there would be single positive hepatocytes for genes 1 or 2, and that, by chance, there would be hepatocytes that are double-positive for both genes. Since this is an expected result, it seems more appropriate for a supplementary figure. Overall, I believe that describing SVP[Rapa] as an enhancer of viral transduction is misleading. SVP[Rapa] was not shown to affect the mechanisms of viral transduction, but did prevent the formation of neutralizing antibodies that may prevent transduction by AAV vectors.

ANSWER: We agree that transduction of hepatocytes is not a “targeted” process and that SVP[Rapa] does not directly enhance viral transduction. We have revised the sentence to read “These results indicate that repeated AAV vector administration enabled by SVP[Rapa] treatment can enhance liver gene transfer in part by increasing the number of hepatocytes that are transduced.” However, to our knowledge it has not been previously demonstrated whether repeated administrations of AAV vectors would transduce the same cells or different cells. Therefore, we believe that it is warranted to keep Figure 3, which shows the data from nonhuman primates. The corresponding mouse data will remain as a supplementary figure.

4. There are no known diseases associated with AAV, however, tolerizing to AAV could have unknown consequences since AAVs do infect numerous people. There were no safety considerations addressed experimentally, nor even discussed in this manuscript.

ANSWER: We thank the reviewer for raising this point. We have added a statement in the discussion that reads “Although AAV is considered to be a non-pathogenic virus and is not associated with any disease, it is unknown if there would be any consequences of inducing immune tolerance to AAV. Further evaluation of safety will have to be addressed in preclinical studies and clinical trials.”

Minor Comments:

1. There is an inconsistency in the amount of rapamycin administered (in Figures 4 and 6 200 µg were administered, in Figure 5, 50 µg were administered). It would be helpful if the authors could comment on the rationale for changing the dose.

ANSWER: As indicated above in the answer to Reviewer 2, higher vector doses are associated with greater immunogenicity. Hence the 50 µg dose of SVP[Rapa] was used with vector doses of 4E11 vg/kg while the 200 µg dose of SVP[Rapa] was used for vector doses of 4E12 vg/kg. We have clarified this in the Methods section.

2. The methods section does not provide sufficient details related to the particle preparation and characterization that would be required for scientists to reproduce this work. There is no description of the loading of rapamycin in the SVP[Rapa] platform or the dose of particles administered in the mouse studies.

ANSWER: We have added additional details on the manufacture of the nanoparticles, particularly in regard to the load of rapamycin and the doses used.

3. In the caption for Figure 1, the description for panels d and e do not clearly describe the positive control of AAV8-hF.IX injected into naïve mice. I am concerned about my interpretation of the results.

ANSWER: We apologize for the omission. The AAV8-hF.IX only group represents a control group of naïve animals that received the same vector dose of 4×10^{12} vg/kg at day 21 without any other reagent. This group show what would be the expected hF.IX transgene expression levels and anti-IgG levels in mice receiving just an AAV vector.

We clarified this point in the figure legend.

4. Figure 1g caption, n value is written as “n=x” instead of actual number of individuals.

ANSWER: We apologize for the omission. We now added the number of mice used in the experiment (n=5).

5. The in vitro neutralizing antibody assay (e.g. Fig 1c, Fig 2d) was not described well enough to describe what is being measured. Where IgG and IgM are clearly defined by fundamental immunology, one must know the assay described in reference 24 to understand the term defined as “neutralizing antibody” here. A brief explanation of the assay in the methods or caption would be clarifying.

ANSWER: We now added more information about the neutralizing antibody assay in the materials and methods section.

“Selected serum samples were also analyzed for anti-AAV neutralizing antibody titer using a previously published in vitro cell-based test²⁵. Briefly, in this assay, serial dilutions of heat-inactivated test samples were mixed with a vector expressing luciferase and incubated for one hour. After incubation, samples were added to cells and residual luciferase expression was measured after 24 hours. The neutralizing titer was determined as the highest sample dilution at which at least 50% inhibition of luciferase expression was measured compared to a non-inhibition control. In this assay, a NAb titer of 1:10 represents the titer of a sample in which after a 10-fold dilution a residual luciferase signal lower than 50% of the non-inhibition control is observed.”

6. The caption for Figure 2f reads, “Plasma hF.IX antigen levels were quantified by ELISA at the indicated time points following liver-directed administration of AAV8-hF.IX vector.” What does liver-directed indicate?

ANSWER: For simplicity, we now removed “liver-directed” from the legend as it can be misleading. By liver directed we meant administration of a vector with a liver-specific transgene promoter.

7. Typographical errors:

o On page 8, “Quantification of hepatocytes IX from...”

ANSWER: Corrected.

o On page 10, “Next, we assessed the ability of SVP[Rapa] to control of primed...”

ANSWER: Corrected.

8. It is unclear what the authors are indicating by “different liver lobes” in Figure 3 caption: “Dual immunofluorescence staining of Gaa and hF.IX in different liver lobes from non-human primates treated”

ANSWER: For each animal wedge biopsies from different lobes of the liver (left, right, quadrate, caudate) were used for immunohistochemistry analysis as outlined in the materials and methods section.

We simplified the figure legends by simply stating “Dual immunofluorescence staining of Gaa and hF.IX in livers from nonhuman primates treated...”.

9. The caption for Figure 4a says n=10, but it looks like only 5 are represented.

ANSWER: We apologize for the typo, we now corrected the figure legend.

10. The text on page 9, and the caption for Figure 4, describes cells from the liver, spleen, and lymph nodes. In these descriptions, please clarify which organs are being referred to, or the sum of all organs, when describing the cell types measured. Also, both spleens and lymph nodes contain germinal centers, so please specify.

ANSWER: We now clarified this point both in the results section and in the figure legend. We apologize for the omission.

11. Page 10: The following text should specify that the described effect was observed specifically in SVP[Rapa], not all SVP-treated animals (which would indicate SVP[empty] as well, “No anti-AAV antibodies were observed in SVP-treated recipient mice.”

ANSWER: Corrected.

12. Figure 6e shows SVP[empty] injection on day 1 after adoptive transfer, but this is not described in the caption or the results section. It is not clear which is accurate, the protocol in the figure, or the caption and the text.

ANSWER: Animal received the vector alone on day 1 after challenge. We now amended the figure to reflect the correct experimental design.

13. Figure 1g shows that tolerance is broken when a subsequent dose of AAV8 is administered with SVP[empty] 93 days after the initial treatment with SVP[Rapa], but Figure 6a shows that tolerance is more durable when a subsequent dose is given at 21 days (although it looks like antibody production is starting to increase at day 41). This is an interesting finding and worth discussion or investigation into the mechanisms of tolerance maintenance.

ANSWER: We apologize for the misunderstanding. We now clarified this point in the figure 1g legend. The animals received either AAV-SEAP alone at day 0 and 93, AAV-

SEAP+SVP[empty] at day 0 and 93, or AAV-SEAP+SVP[Rapa] at day 0 and 93. Thus, the experiment shows that the effect is durable as long as SVP[Rapa] are given with the vector. The figure 1f legends now reads as follows:

“Three groups of C57BL/6 male mice (n=5/group) were prime-boosted (days 0 and 93, arrows) with 5×10^{11} vg/kg of AAV8-SEAP alone, or AAV8-SEAP mixed with SVP[Rapa] (50 μ g), or AAV8-SEAP mixed with SVP[Empty] control.”

References:

- 1 Mauri, C. & Menon, M. Human regulatory B cells in health and disease: therapeutic potential. *J Clin Invest* **127**, 772-779, doi:10.1172/JCI85113 (2017).
- 2 Kishimoto, T. K. *et al.* Improving the efficacy and safety of biologic drugs with tolerogenic nanoparticles. *Nat Nanotechnol* **11**, 890-899, doi:10.1038/nnano.2016.135 (2016).
- 3 Sands, E., Kivitz, A. J., DeHaan, W., Johnston, L. & Kishimoto, T. K. Initial phase 2 clinical data of SEL-212 in symptomatic gout patients: monthly dosing of a pegylated uricase (pegsiticase) with SVP-rapamycin enables sustained reduction of serum uric acid levels by mitigating formation of anti-drug antibodies *Arthritis Rheumatol* **69** (2017).
- 4 Ito, T. *et al.* A convenient enzyme-linked immunosorbent assay for rapid screening of anti-adenovirus neutralizing antibodies. *Ann Clin Biochem* **46**, 508-510, doi:10.1258/acb.2009.009077 (2009).
- 5 Unzu, C. *et al.* Transient and intensive pharmacological immunosuppression fails to improve AAV-based liver gene transfer in non-human primates. *Journal of translational medicine* **10**, 122, doi:10.1186/1479-5876-10-122 (2012).
- 6 Mingozzi, F. *et al.* Modulation of tolerance to the transgene product in a nonhuman primate model of AAV-mediated gene transfer to liver. *Blood* **110**, 2334-2341, doi:10.1182/blood-2007-03-080093 [pii] 10.1182/blood-2007-03-080093 (2007).
- 7 Jiang, H. *et al.* Effects of transient immunosuppression on adenoassociated, virus-mediated, liver-directed gene transfer in rhesus macaques and implications for human gene therapy. *Blood* **108**, 3321-3328, doi:10.1182/blood-2006-04-017913 [pii] 10.1182/blood-2006-04-017913 (2006).
- 8 Fagioli, S., Daina, E., D'Antiga, L., Colledan, M. & Remuzzi, G. Monogenic diseases that can be cured by liver transplantation. *J Hepatol* **59**, 595-612, doi:10.1016/j.jhep.2013.04.004 (2013).
- 9 Bortolussi, G. *et al.* Life-long correction of hyperbilirubinemia with a neonatal liver-specific AAV-mediated gene transfer in a lethal mouse model of Crigler-Najjar Syndrome. *Human gene therapy* **25**, 844-855, doi:10.1089/hum.2013.233 (2014).
- 10 Rangarajan, S. *et al.* AAV5-Factor VIII Gene Transfer in Severe Hemophilia A. *N Engl J Med* **377**, 2519-2530, doi:10.1056/NEJMoa1708483 (2017).
- 11 Le Guiner, C. *et al.* Long-term microdystrophin gene therapy is effective in a canine model of Duchenne muscular dystrophy. *Nat Commun* **8**, 16105, doi:10.1038/ncomms16105 (2017).
- 12 Elverman, M. *et al.* Long-term effects of systemic gene therapy in a canine model of myotubular myopathy. *Muscle & nerve* **56**, 943-953, doi:10.1002/mus.25658 (2017).

- 13 Greig, J. A. *et al.* Intramuscular administration of AAV overcomes pre-existing neutralizing antibodies in rhesus macaques. *Vaccine* **34**, 6323-6329, doi:10.1016/j.vaccine.2016.10.053 (2016).
- 14 Amado, D. *et al.* Safety and efficacy of subretinal readministration of a viral vector in large animals to treat congenital blindness. *Science translational medicine* **2**, 21ra16, doi:10.1126/scitranslmed.3000659 (2010).
- 15 Bennett, J. *et al.* AAV2 gene therapy readministration in three adults with congenital blindness. *Science translational medicine* **4**, 120ra115, doi:10.1126/scitranslmed.3002865 (2012).
- 16 Manno, C. S. *et al.* AAV-mediated factor IX gene transfer to skeletal muscle in patients with severe hemophilia B. *Blood* **101**, 2963-2972, doi:10.1182/blood-2002-10-3296 (2003).
- 17 Chicoine, L. G. *et al.* Plasmapheresis eliminates the negative impact of AAV antibodies on microdystrophin gene expression following vascular delivery. *Molecular therapy : the journal of the American Society of Gene Therapy* **22**, 338-347, doi:10.1038/mt.2013.244 (2014).
- 18 Monteilhet, V. *et al.* A 10 patient case report on the impact of plasmapheresis upon neutralizing factors against adeno-associated virus (AAV) types 1, 2, 6, and 8. *Molecular therapy : the journal of the American Society of Gene Therapy* **19**, 2084-2091, doi:10.1038/mt.2011.108 (2011).
- 19 Majowicz, A. *et al.* Successful Repeated Hepatic Gene Delivery in Mice and Non-human Primates Achieved by Sequential Administration of AAV5(ch) and AAV1. *Molecular therapy : the journal of the American Society of Gene Therapy* **25**, 1831-1842, doi:10.1016/j.ymthe.2017.05.003 (2017).
- 20 Boutin, S. *et al.* Prevalence of serum IgG and neutralizing factors against adeno-associated virus (AAV) types 1, 2, 5, 6, 8, and 9 in the healthy population: implications for gene therapy using AAV vectors. *Human gene therapy* **21**, 704-712, doi:10.1089/hum.2009.182 (2010).
- 21 Mingozi, F. *et al.* Overcoming preexisting humoral immunity to AAV using capsid decoys. *Science translational medicine* **5**, 194ra192, doi:10.1126/scitranslmed.3005795 (2013).
- 22 Gao, G. *et al.* Adeno-associated virus-mediated gene transfer to nonhuman primate liver can elicit destructive transgene-specific T cell responses. *Human gene therapy* **20**, 930-942, doi:10.1089/hum.2009.060 (2009).
- 23 Ronzitti, G. *et al.* A translationally optimized AAV-UGT1A1 vector drives safe and long-lasting correction of Crigler-Najjar syndrome. *Mol Ther Methods Clin Dev* **3**, 16049, doi:10.1038/mtm.2016.49 (2016).
- 24 Puzzo, F. *et al.* Rescue of Pompe disease in mice by AAV-mediated liver delivery of secretable acid alpha-glucosidase. *Science translational medicine* **9**, doi:10.1126/scitranslmed.aam6375 (2017).
- 25 Meliani, A. *et al.* Determination of anti-adeno-associated virus vector neutralizing antibody titer with an in vitro reporter system. *Hum Gene Ther Methods* **26**, 45-53, doi:10.1089/hgtb.2015.037 (2015).

REVIEWERS' COMMENTS:

Reviewer #1 (Remarks to the Author):

All major issues have been addressed and this manuscript should be published as soon as feasible.

Reviewer #2 (Remarks to the Author):

I thank the authors for their detailed and considered responses to my comments. I am satisfied with the revisions and explanations given.

Reviewer #3 (Remarks to the Author):

Though this manuscript does not introduce a novel immunomodulatory platform (SVP[Rapa]), it does introduce a new indication of the platform as it may be applied to enabling multiple doses of adenovirus for gene therapy.

The authors claim that concurrent delivery of AAV8 with SVP[Rapa] was more effective than AAV8 dosed 24 hours later, but it appears that both treatment regimens had an effect up to day 33. This seems to indicate that there is a window of effective antigen-selective immunomodulation greater than 24 hours, but less than 3 weeks, from SVP[Rapa] treatment. Therefore, neither Figure 6 or the figure above demonstrate that SVP[Rapa] is not globally immunosuppressive. I recommend that the authors do not make this claim.

The methods section has been improved. However, the description "SVP[Rapa] contained ~10% rapamycin load" is vague. It is recommended that the authors use more descriptive words or units (e.g. % wt/wt or µg/mg).

REVIEWERS' COMMENTS:

Reviewer #1 (Remarks to the Author):

All major issues have been addressed and this manuscript should be published as soon as feasible.

ANSWER: We would like to thank again the Reviewer for helping us improve the quality of our manuscript

Reviewer #2 (Remarks to the Author):

I thank the authors for their detailed and considered responses to my comments. I am satisfied with the revisions and explanations given.

ANSWER: We would like to thank again the Reviewer for helping us improve the quality of our manuscript

Reviewer #3 (Remarks to the Author):

Though this manuscript does not introduce a novel immunomodulatory platform (SVP[Rapa]), it does introduce a new indication of the platform as it may be applied to enabling multiple doses of adenovirus for gene therapy.

ANSWER: We thank the Reviewer for acknowledging the relevance of our work. Indeed, addressing AAV vector immunogenicity and enabling vector redosing remain major challenges for the field of in vivo gene therapy.

The authors claim that concurrent delivery of AAV8 with SVP[Rapa] was more effective than AAV8 dosed 24 hours later, but it appears that both treatment regimens had an effect up to day 33. This seems to indicate that there is a window of effective antigen-selective immunomodulation greater than 24 hours, but less than 3 weeks, from SVP[Rapa] treatment. Therefore, neither Figure 6 or the figure above demonstrate that SVP[Rapa] is not globally immunosuppressive. I recommend that the authors do not make this claim.

ANSWER: We understand the Reviewer's concern. We now edited the manuscript by removing the term "global" in the Results section and also edited the text of the Discussion section, which now reads as follows:

"Because of the need of both the need for SVP[Rapa] to be co-administered with the antigen, in order to guarantee uptake from the same subset of antigen presenting cells^{18,21}, it is less

likely that SVP[Rapa] will lead to global immunosuppression, although future studies are needed to carefully address this point.”

The methods section has been improved. However, the description “SVP[Rapa] contained ~10% rapamycin load” is vague. It is recommended that the authors use more descriptive words or units (e.g. % wt/wt or µg/mg).

ANSWER: Rapamycin content was measured by first transferring the nanoparticles into an organic solvent, followed by analysis via reverse-phase HPLC using a water/acetonitrile gradient containing 0.1% TFA. Typically, SVP[Rapa] contained ~10% rapamycin load; in a given preparation of SVP[Rapa], the concentration of rapamycin was adjusted to 2 mg/ml. Nanoparticle size was ~200 nm as determined by dynamic light scattering. Each SVP[Rapa] treatment consisted of 50 µg or 200 µg of rapamycin as indicated in figure legend.